# Synthesis and Antiproliferative Activity of 2,4,6,7-Tetrasubstituted-2*H*-pyrazolo[4,3-*c*]pyridines

**DOI:** 10.3390/molecules26216747

**Published:** 2021-11-08

**Authors:** Beatričė Razmienė, Eva Řezníčková, Vaida Dambrauskienė, Radek Ostruszka, Martin Kubala, Asta Žukauskaitė, Vladimír Kryštof, Algirdas Šačkus, Eglė Arbačiauskienė

**Affiliations:** 1Department of Organic Chemistry, Kaunas University of Technology, Radvilėnų pl. 19, LT-50254 Kaunas, Lithuania; beatrice.razmiene@ktu.lt (B.R.); laukaityte.vaida@gmail.com (V.D.); algirdas.sackus@ktu.lt (A.Š.); 2Institute of Synthetic Chemistry, Kaunas University of Technology, K. Baršausko g. 59, LT-51423 Kaunas, Lithuania; 3Department of Experimental Biology, Palacký University, Šlechtitelů 27, CZ-78371 Olomouc, Czech Republic; eva.reznickova@upol.cz (E.Ř.); vladimir.krystof@upol.cz (V.K.); 4Department of Experimental Physics, Faculty of Science, Palacký University, 17. Listopadu 12, CZ-77146 Olomouc, Czech Republic; radek.ostruszka@upol.cz (R.O.); martin.kubala@upol.cz (M.K.); 5Department of Chemical Biology, Palacký University, Šlechtitelů 27, CZ-78371 Olomouc, Czech Republic; 6Institute of Molecular and Translational Medicine, Faculty of Medicine and Dentistry, Palacký University, Hněvotínská 5, CZ-77900 Olomouc, Czech Republic

**Keywords:** antiproliferation, cell death, cross-coupling, cycloiodination, pyrazole, pyridine

## Abstract

A library of 2,4,6,7-tetrasubstituted-2*H*-pyrazolo[4,3-*c*]pyridines was prepared from easily accessible 1-phenyl-3-(2-phenylethynyl)-1*H*-pyrazole-4-carbaldehyde via an iodine-mediated electrophilic cyclization of intermediate 4-(azidomethyl)-1-phenyl-3-(phenylethynyl)-1*H*-pyrazoles to 7-iodo-2,6-diphenyl-2*H*-pyrazolo[4,3-*c*]pyridines followed by Suzuki cross-couplings with various boronic acids and alkylation reactions. The compounds were evaluated for their antiproliferative activity against K562, MV4-11, and MCF-7 cancer cell lines. The most potent compounds displayed low micromolar GI_50_ values. 4-(2,6-Diphenyl-2*H*-pyrazolo[4,3-*c*]pyridin-7-yl)phenol proved to be the most active**,** induced poly(ADP-ribose) polymerase 1 (PARP-1) cleavage, activated the initiator enzyme of apoptotic cascade caspase 9, induced a fragmentation of microtubule-associated protein 1-light chain 3 (LC3), and reduced the expression levels of proliferating cell nuclear antigen (PCNA). The obtained results suggest a complex action of 4-(2,6-diphenyl-2*H*-pyrazolo[4,3-*c*]pyridin-7-yl)phenol that combines antiproliferative effects with the induction of cell death. Moreover, investigations of the fluorescence properties of the final compounds revealed 7-(4-methoxyphenyl)-2,6-diphenyl-2*H*-pyrazolo[4,3-*c*]pyridine as the most potent pH indicator that enables both fluorescence intensity-based and ratiometric pH sensing.

## 1. Introduction

Despite being rarely found in nature, presumably due to the difficulty of forming N–N bonds in living organisms, naturally occurring pyrazoles are prominent in laboratories due to their vast variety of biological activities [1]. In current medicinal chemistry, the incorporation of a pyrazole nucleus is a common practice to develop new drug-like molecules with anti-cancer, anti-diabetic, anti-viral, anti-inflammatory, anti-bacterial, anti-fungal, anti-neurodegenerative, anti-tubercular, anthelmintic, antimalarial, and photosensitizing properties [2,3,4,5,6,7,8,9,10], among others, thus giving rise to a great number of approved therapeutics [11]. Besides numerous biological activities, pyrazoles have also been documented to possess dyeing and fluorescence properties [12,13,14,15,16], and some of them can be used as colorimetric or fluorescent probes for sensing small molecules, ions, or pH [17,18,19,20,21,22,23,24,25,26,27,28], which may have applications in in vivo imaging [29,30,31]. Pyrazolopyridines are among the most studied condensed pyrazole systems in organic and pharmaceutical chemistry (Figure 1). For instance, 6-(3,5-dimethoxyphenyl)-3-(4-fluorophenyl)-1*H*-pyrazolo[3,4-*b*]pyridine (APcK110) is an extensively researched Kit kinase inhibitor [32,33,34]. More recently, various 1*H*-pyrazolo[3,4-*b*]pyridine derivatives were reported as potent ALK-L1196M [35] and CDK8 inhibitors [36], PPARα agonists [37], and antimicrobial, anti-quorum-sensing, and anticancer agents [38], while 3-amino-1*H*-pyrazolo[3,4-*b*]pyridine core was identified as a novel scaffold for MELK kinase inhibitors [39]. 2-{[2-(1*H*-Pyrazolo[3,4-*c*]pyridin-3-yl)-6-(trifluoromethyl)pyrimidin-4-yl]amino}ethanol is a bacterial DNA ligase inhibitor [40], several compounds bearing 1*H*-pyrazolo[4,3-*b*]pyridin-3-amine scaffold act as positive allosteric modulators of the metabotropic glutamate receptor 4 (mGlu_4_) [41,42], 3-phenylpyrazolo[3,4-*c*]pyridines were reported to possess antiproliferative activity [43], and 1-(4-methoxybenzyl)-7-(4-methylpiperazin-1-yl)-*N*-[4-(4-methylpiperazin-1-yl)phenyl]-3-phenyl-1*H*-pyrazolo[3,4-*c*]pyridin-5-amine was suggested as potential angiogenesis inhibitor [44]. Among biologically active pyrazolo[4,3-*c*]pyridines, 3-amino-2-phenyl-2*H*-pyrazolo[4,3-*c*]pyridine-4,6-diol has shown inhibitory activity against p90 ribosomal S6 kinases 2 (RSK2) [45], while 3-aminopyrazolopyridinone derivatives were demonstrated to exhibit moderate inhibitory potency against CK1d, p38a, and aurora A kinases [46].

In a continuation of our work devoted to the preparation and study of the properties of various condensed and aryl coupled pyrazole derivatives [47,48,49,50,51,52,53,54,55], we recently reported a structure–activity relationship study on 2,4,6-trisubstituted-2*H*-pyrazolo[4,3-*c*]pyridines, several of which displayed good anticancer activity in vitro through arresting cell cycle in mitosis and the induction of apoptosis [54]. Inspired by these results, in the current work, we prepared a library of 2,4,6,7-tetrasubstituted-2*H*-pyrazolo[4,3-*c*]pyridines and examined the influence of an additional substituent at the 7-position on the biological and optical properties of the compounds.

## 2. Results and Discussion

### 2.1. Chemistry

1-Phenyl-3-(2-phenylethynyl)-1*H*-pyrazole-4-carbaldehyde **2**, which served as a starting material in this study, was prepared via a multi-step synthetic route from 1-phenyl-1*H*-pyrazol-3-ol **1** in accordance with a previously published procedure [56] (Appendix A). Then, primary alcohol **3** was obtained via the reduction of an aldehyde group [57] (Figure 1). Sodium borohydride was chosen as a reducing agent, and the reaction was carried out in methanol at 0 °C under an argon atmosphere. The reaction mixture was protonated with an aqueous ammonium chloride solution to create primary alcohol **3** from the intermediate complex. For the synthesis of secondary alcohols **4**–**7**, carbaldehyde **2** was dissolved in THF and reacted with an appropriate alkyl or arylmagnesium halogenide at room temperature by adopting a previously reported procedure [58]. The reaction was carried out under an argon atmosphere with a dry solvent due to the sensitivity of Grignard reagents to air and moisture [59]. Although it is known that this kind of secondary alcohol might be unstable [51], all of them were successfully purified by column chromatography, and their structures were determined with spectroscopic data.

The obtained alcohols **3**–**7** were further converted into azides **9**–**12**, respectively. Many methods have been developed for such a transformation, including Mitsunobu-type displacements [60,61], two-step procedures that involve a halogenated [62] or mesylated intermediate [63], the one-pot halogenation-azidation of alcohols [64], reactions with phosphitine intermediates [65], *N*-methyl-2-pyrolidone hydrosulphate, and trimethylsilylazide (TMSN_3_) [51]. The latter method was chosen for the synthesis, and the reactions were performed in DCM with a catalytic amount of boron trifluoride diethyl etherate (BF_3_∙Et_2_O). The reactions were carried out at room temperature under an argon atmosphere and with a dry solvent in order to protect both the boron trifluoride and TMSN_3_ from moisture (Figure 1). Conversion was completed in 30 minutes, and the reaction products **8**–**12** were furnished in 50–93% yields.

The newly synthesized azides **8**–**12** were further used to form the pyrazolo[4,3-*c*]pyridine core with iodine in the 7-position by adopting electrophilic substitution reaction conditions that were previously used to obtain 1,3,4-trisubstituted isoquinolines from 2-alkynyl benzyl azides [66]. Namely, azides **8**–**12** were dissolved in DCM and treated with iodine and a proper base (Figure 1). Five equivalents of K_3_PO_4_ were used for the primary azides **8** and **12**, while one equivalent of NaHCO_3_ was used for the secondary azides **9**–**11**. The reactions were carried out at room temperature in the dark for 12 h, furnishing compounds **13**–**17** in 70–88% yields. An attempt to make use of a weaker base NaHCO_3_ for the reaction with the primary azide **6** led to the formation of the dehalogenated side product 2,6-diphenyl-2*H*-pyrazolo[4,3-*c*]pyridine, which resulted in a troublesome purification and a lower yield of the target product.

The obtained 7-iodo-2,6-diphenyl-2*H*-pyrazolo[4,3-*c*]pyridines **13**–**17** were further used in palladium-catalysed Suzuki–Miyaura cross-coupling reaction (Figure 2) by adopting a previously reported procedure [67]. Namely, aromatic boronic acids were reacted with compounds **13**–**17** using palladium acetate as a catalyst and Cs_2_CO_3_ as a base in an aqueous ethanol solution under an argon atmosphere. To ensure a short reaction time, cross-coupling reactions were carried out under microwave irradiation, thus giving rise to compounds **18**–**37**. Compounds **18** and **20**–**37** (Figure 2) were obtained in fair to excellent yields (48–96%), but the full cross-coupling conversion of **13** using 2-methoxyphenyl boronic acid could not be achieved, resulting in a lower yield of compound **19**.

4-ethyl-7-(4-hydroxyphenyl)-2,6-diphenyl-2*H*-pyrazolo[4,3-*c*]pyridine **37** was further subjected to hydroxyl group alkylation reactions with ethyl, propyl, and isopropyl iodides. As a result, 7-(4-alkoxyphenyl)-4-ethyl-2,6-diphenyl-2*H*-pyrazolo[4,3-*c*]pyridines **40**–**42** were obtained in high yields (80–97%) (Figure 3).

### 2.2. Optical Properties

The fluorescence properties of all final compounds **18**–**42** were first investigated in THF, with the excitation wavelength *λ*_ex_ being set to 350 nm (Appendix A). The emission maxima *λ*_em_ of all the compounds were located in the 437–487 nm range, which corresponds to the blue part of the visible light spectrum. A polar 4-hydroxyphenyl substituent at the 7-position bearing compounds **23**, **29,** and **37**, as well as derivatives **31**–**32**, **38**, **40**–**42** (all of which bear 4-alkoxyphenyl substituents at the 7-position and ethyl or isopropyl substituents at the 4-position), possessed the most pronounced fluorescence properties. Namely, the Stokes shifts for these compounds were in 199–205 nm range, and the quantum yield reached approximately 60–85%.

Intracellular pH plays an important role in many biological processes, and its changes from normal to abnormal levels can lead to cellular dysfunction, various diseases, and a decrease in physical performance [68]. pH-sensitive fluorescent indicators enable the precise measurement of intracellular pH, which consequently provides valuable information about ongoing physiological and pathological processes at the cellular and sub-cellular levels [69]. To assess whether the fluorescence properties of the prepared compounds are pH-dependent, they were all analysed in pH 5, 7, and 9 buffers with the excitation wavelength *λ*_ex_ set to 360 nm (Appendix A). The quantum yield of compounds **18**, **24,** and **30**, all of which bear phenyl substituent at the 7-position, increased at acidic pH without substantial shifts in emission maxima, which were observed to be in the 435–447 nm range. The further analysis of compound **18** in a range of pH 2–11 buffers revealed a gradual decrease in fluorescence intensity with the increase of pH (Appendix A). On the other hand, the quantum yield of 2-methoxyphenyl or 4-methoxyphenyl substituent at the 7-position possessing compounds **19**, **21**, **25**, **27,** and **31** was higher in basic pH; moreover, in the case of 4-methoxyphenyl substituent at the 7-position bearing compounds **21**, **27,** and **31**, an acidic pH caused the red shift of the emission maxima. For instance, in the case of compound **21**, the emission spectrum was found to be composed of two partly overlapping bands (Appendix A). The short-wavelength part is pH-sensitive. It is dominant in the basic environment, and decreasing the pH from 11 to 6 caused a decrease in fluorescence intensity without a shift in emission maxima, which was maintained at 458 nm. After a further decrease of pH, it was the long-wavelength band that became more dominant, which was manifested as a gradual shift of emission maxima to 519 nm. Any other 4-alkoxyphenyl substituent at the 7-position bearing compounds only exhibited a drop of quantum yield at acidic pH without the shift of the emission maxima. A polar 4-hydroxyphenyl substituent at the 7-position bearing compounds **23**, **29,** and **37**, which possessed the highest quantum yields in THF, had the lowest quantum yields of up to 0.3% in an aqueous solution. It is well known that hydroxyphenyl groups are sensitive to photochemical reactions [70]. Typically, their pK_a_ drops from ~10 in the ground state to ~3 in the excited state, and excited-state proton transfer reactions are common in aqueous solutions. For our molecules, these reactions resulted in fluorescence quenching. Our preliminary observations suggested that most of the compounds, except for the polar derivatives **23**, **29,** and **37**, could be of potential interest as pH indicators. Considering the approximately 5-fold quantum yield increase and 40 nm blue shift of the emission spectrum maximum when moving from pH 5 to 9, the compound **21** seems to be the best pH indicator from the set of examined molecules, enabling both fluorescence-intensity-based and ratiometric pH sensing.

### 2.3. Biology

Synthesized 2*H*-pyrazolo[4,3-*c*]pyridines **18**–**42** were evaluated for their antiproliferative activity against three human cancer cell lines, i.e., MV4-11 (biphenotypic B myelomonocytic leukaemia cells), K562 (chronic myeloid leukaemia cells), and MCF-7 (human breast cancer cells) (Table 1). In agreement with our previous observations [54], increasing the bulkiness of the substituent at the 4-position gradually reduced or completely abolished the activity of 2*H*-pyrazolo[4,3-*c*]pyridine derivatives. Namely, while methyl-substituted compounds **24**–**29** possessed lower antiproliferative values than their unsubstituted counterparts **18**–**23**, isopropyl-, ethyl-, and phenyl-substituted compounds were mostly non-active up to the tested 10 µM concentration. Overall, a polar 4-hydroxyphenyl substituent at the 7-position bearing the 4-unsubstituted derivative **23** proved to be the most potent.

Subsequently, the effects of the most active compound **23**, its less active 4-substituted analogues **29** and **37**, and derivative **21** were studied on K562 leukemic cells. Asynchronously growing K562 cells were treated with 10 µM concentrations of selected compounds for 24, 48, and 72 h and analysed using immunoblotting and flow cytometry (Figure 2). Immunoblotting revealed that 48 h treatment with the most potent compound **23** was sufficient for the induction of poly(ADP-ribose) polymerase 1 (PARP-1) cleavage [71] and the activation of initiator enzyme of apoptotic cascade caspase 9 [72]. Interestingly, in addition to the clear pro-apoptotic effects, we also observed the time-dependent fragmentation of microtubule-associated protein 1-light chain 3 (LC3), which has appeared during autophagy [73]. Similar outcomes with lower efficiencies were observed in all tested compounds. In addition to cell-death-related proteins, the expression levels of proliferating cell nuclear antigen (PCNA), which plays a key role in DNA replication [74], were analysed. The results revealed that all studied compounds reduced the levels of PCNA time-dependently, with the most pronounced effect observed for compounds **23** and **29**. To independently support this observation, immunoblotting was complemented with the flow cytometric analysis of bromodeoxyuridine (BrdU) incorporation, which allowed us to recognize replicating BrdU-positive cells in the population [75] (Figure 2B). In control samples the number of proliferating cells came up to 40%, but the 10 µM treatment with tested compounds **21**, **23**, **29,** and **37** reduced the proportion of actively proliferating BrdU-positive cells in up to approximately 10% for the most active compounds **23** and **29**. Overall, the obtained results suggest the complex action of the compounds, combining antiproliferative effects with the induction of cell death.

## 3. Materials and Methods

### 3.1. General

All chemicals and solvents were purchased from commercial suppliers and used without further purification unless otherwise specified. The ^1^H, ^13^C, and ^15^N NMR spectra were recorded in CDCl_3_ or DMSO-*d*_6_ solutions at 25 °C on either a Bruker Avance III 700 (700 MHz for ^1^H, 176 MHz for ^13^C, and 71 MHz for ^15^N) spectrometer equipped with a 5 mm TCI ^1^H-^13^C/^15^N/D z-gradient cryoprobe or a Jeol ECA-500 (500 MHz for ^1^H and 126 MHz for ^13^C) spectrometer equipped with a 5 mm Royal probe. The chemical shifts, expressed in ppm, were relative to tetramethylsilane (TMS). The ^15^N NMR spectra were referenced to neat, external nitromethane (coaxial capillary). The ^19^F NMR spectra (376 MHz) were obtained on a Bruker Avance III 400 instrument using C_6_F_6_ as an internal standard. FT-IR spectra were collected using the ATR method on a Bruker Vertex 70v spectrometer with an integrated Platinum ATR accessory or on a Bruker Tensor 27 spectrometer in KBr pellets. The melting points of crystalline compounds were determined in open capillary tubes with a Buchi M 565 apparatus (temperature gradient: 2 °C/min) and are uncorrected. Mass spectra were recorded on Q-TOF MICRO spectrometer (Waters), analyses were performed in the positive (ESI^+^) mode, and molecular ions were recorded in [M + H]^+^ forms. High-resolution mass spectrometry (HRMS) spectra were obtained in the ESI mode on a Bruker MicrOTOF-Q III spectrometer. All reactions were performed in oven-dried flasks under an argon atmosphere with magnetic stirring. Reaction progress was monitored by TLC analysis on Macherey-Nagel™ ALUGRAM^®^ Xtra SIL G/UV254 plates. TLC plates were visualized with UV light (wavelengths of 254 and 365 nm) or iodine vapour. Compounds were purified by flash chromatography in a glass column (stationary phase of silica gel, high-purity grade of 9385, pore size of 60 Å, particle size of 230–400 mesh, supplied by Sigma-Aldrich). ^1^H, ^13^C, and ^1^H-^15^N HMBC NMR spectra, as well as the HRMS data of new compounds, are provided in Appendix A.

### 3.2. Chemistry

#### 3.2.1. Procedure for the Synthesis of [1-Phenyl-3-(phenylethynyl)-1*H*-pyrazol-4-yl]methanol **3**

1-Phenyl-3-(2-phenylethynyl)-1*H*-pyrazole-4-carbaldehyde **3** (560 mg, 2.06 mmol) was dissolved in MeOH (12 mL), and the solution was cooled to 0 °C. Subsequently, NaBH_4_ (156 mg, 4.12 mmol) was added under an argon atmosphere, and the mixture was stirred for 30 min. Upon completion (monitored by TLC), the reaction mixture was diluted with a saturated aqueous NH_4_Cl solution (20 mL) and extracted with EtOAc (3 × 30 mL). The combined organic layers were dried over anhydrous Na_2_SO_4_, filtered, and evaporated under reduced pressure. The residue was purified by column chromatography (EtOAc/Hex, 1:3 *v*/*v*). Yield: 530 mg (94%), white crystalline solid, mp = 114–115 °C, R*_f_* = 0.18 (EtOAc/Hex, 1:3 *v*/*v*). ^1^H NMR (700 MHz, CDCl_3_): δ 2.12 (1H, s, OH), 4.77 (2H, d, *J* = 7.0 Hz, C*H*_2_OH), 7.28–7.32 (1H, m, NPh 4-H), 7.33–7.38 (3H, m, CPh 3,4,5), 7.42–7.46 (2H, m, NPh 3,5-H), 7.55–7.59 (2H, m, CPh 2,6-H), 7.66–7.72 (2H, m, NPh 2,6-H), 7.95 (1H, s, 5-H). ^13^C NMR (176 MHz, CDCl_3_): δ 55.9 (*C*H_2_OH), 80.3 (*C*≡CPh), 93.7 (C≡*C*Ph), 119.4 (NPh C-2,6), 122.5 (C-4), 126.4 (C-5 and CPh C-1), 127.1 (NPh C-4), 128.5 (CPh C-3,5), 128.9 (CPh C-4), 129.6 (NPh C-3,5), 131.9 (CPh C-2,6), 135.6 (C-3), 139.7 (NPh C-1). ^15^N NMR (71 MHz, CDCl_3_): δ −163.5 (N-1), N-2 not found. IR (ν, cm^−1^): 3373 (OH), 3126, 3066, 3056 (CH_arom_), 2920, 2864 (CH_aliph_), 1599, 1502, 1335, 1217 (C=C, C=N, C–N), 1063, 1014 (CH_2_-OH), 749, 686 (CH=CH of mono- and disubstituted benzenes). MS (ES^+^): *m*/*z* (%): 275 ([M + H]^+^, 100). HRMS (ESI) for C_18_H_14_N_2_ONa ([M + Na]^+^): requires 297.0998 and found 297.0988.

#### 3.2.2. General Procedure (A) for the Synthesis of Alcohols **4**–**7**

1-Phenyl-3-(2-phenylethynyl)-1*H*-pyrazole-4-carbaldehyde **2** (1 equivalent) was dissolved in dry THF under an argon atmosphere. Subsequently, an appropriate Grignard reagent (1.2 equivalents) was added, and the mixture was stirred at room temperature for 10 min. Upon completion (monitored by TLC), the reaction mixture was diluted with water (20 mL) and extracted with EtOAc (3 × 20 mL). The combined organic layers were dried over anhydrous Na_2_SO_4_, filtered, and evaporated under reduced pressure. The residue was purified by column chromatography.

##### 1-[1-Phenyl-3-(phenylethynyl)-1*H*-pyrazol-4-yl]ethanol-1-ol **4**

1-[1-Phenyl-3-(phenylethynyl)-1*H*-pyrazol-4-yl]ethanol-1-ol **4** was prepared in accordance with general procedure (A) from 1-phenyl-3-(2-phenylethynyl)-1*H*-pyrazole-4-carbaldehyde **2** (350 mg, 1.287 mmol) and MeMgCl (0.52 mL, 1.56 mmol) in THF (4 mL). The desired compound was purified by column chromatography (EtOAc/Hex, 1:4 *v*/*v*). Yield: 286 mg (70%), colourless liquid, R*_f_* = 0.23 (EtOAc/Hex, 1:5 *v*/*v*). ^1^H NMR (700 MHz, CDCl_3_): δ 1.64 (3H, d, *J* = 6.5 Hz, CH_3_), 2.18 (1H, s, OH), 5.13 (1H, q, *J* = 6.5 Hz, CH), 7.29–7.1 (1H, m, NPh 4-H), 7.34–7.38 (3H, m, CPh 3,4,5-H), 7.43–7.47 (2H, m, NPh 3,5-H), 7.55–7.59 (2H, m, CPh 2,6-H), 7.67–7.72 (2H, m, NPh, 2,6-H), 7.92 (1H, s, 5-H). ^13^C NMR (176 MHz, CDCl_3_): δ 23.8 (CH_3_), 62.7 (CH) 80.6 (*C*≡CPh), 93.8 (C≡*C*Ph), 119.1 (NPh C-2,6), 122.4 (CPh C-1), 124.4 (C-5), 126.9 (NPh C-4), 128.4 (CPh C-3,5), 128.7 (CPh C-4), 129.4 (NPh C-3,5), 131.5 (C-1), 131.7 (CPh C-2,6), 134.2 (C-2), 139.6 (NPh C-1). ^15^N NMR (71 MHz, CDCl_3_): δ −75.4 (N-2), −164.4 (N-1). MS (ES^+^): *m*/*z* (%): 275 ([M + H]^+^, 96).

##### 1-[1-Phenyl-3-(phenylethynyl)-1*H*-pyrazol-4-yl]propan-1-ol **5**

1-[1-Phenyl-3-(phenylethynyl)-1*H*-pyrazol-4-yl]propan-1-ol **5** was prepared in accordance with general procedure (A) from 1-phenyl-3-(2-phenylethynyl)-1*H*-pyrazole-4-carbaldehyde **2** (100 mg, 0.37 mmol) and EtMgBr (0.15 mL, 0.44 mmol) in THF (2 mL). The desired compound was purified by column chromatography (EtOAc/Hex, 1:3 *v*/*v*). Yield: 100 mg (90%), yellowish crystalline solid, mp = 87–88 °C, R*_f_* = 0.28 (EtOAc/Hex, 1:3 *v*/*v*). ^1^H NMR (700 MHz, CDCl_3_): δ 1.03 (3H, t, *J* = 7.7 Hz, CH_3_), 1.93–1.99 (2H, m, CH_2_), 2.22 (1H, br s, OH), 4.87 (1H, t, *J* = 6.5 Hz, CH), 7.29–7.32 (1H, m, NPh 4-H), 7.34–7.38 (3H, m, CPh 3,4,5-H), 7.43–7.46 (2H, m, NPh 3,5-H), 7.55–7.59 (2H, m, CPh 2,6-H), 7.68–7.72 (2H, m, NPh 2,6-H), 7.91 (1H, s, 5-H). ^13^C NMR (176 MHz, CDCl_3_): δ 10.0 (CH_3_), 30.8 (CH_2_), 68.1 (*C*HOH), 80.7 (*C*≡CPh), 93.6 (C≡*C*Ph), 119.1 (NPh C-2,6), 122.5 (CPh C-4), 124.8 (C-5), 126.9 (NPh C-4), 128.4 (CPh C-3,5), 128.7 (CPh C-4), 129.4 (NPh C-3,5), 130.2 (C-4), 131.6 (CPh C-2,6), 134.5 (C-3), 139.6 (NPh C-1). ^15^N NMR (71 MHz, CDCl_3_): δ −75.8 (N-2), −164.1 (N-1). IR (KBr, ν, cm^−1^): 3441 (OH), 3056 (CH_arom_), 2960, 2874 (CH_aliph_), 1596, 1502, 1335, 1212 (C=C, C=N, C–N), 1063, 109 (CH_2_-OH), 755, 691 (CH=CH of mono- and disubstituted benzenes). MS (ES^+^): *m*/*z* (%): 303 ([M+H]^+^, 98). HRMS (ESI) for C_20_H_18_N_2_ONa ([M + Na]^+^): requires 325.1311 and found 325.1311.

##### 2-Methyl-1-[1-phenyl-3-(phenylethynyl)-1*H*-pyrazol-4-yl]propan-1-ol **6**

2-Methyl-1-[1-phenyl-3-(phenylethynyl)-1*H*-pyrazol-4-yl]propan-1-ol **6** was prepared in accordance with general procedure (A) from 1-phenyl-3-(2-phenylethynyl)-1*H*-pyrazole-4-carbaldehyde **2** (350 mg, 1.287 mmol) and iPrMgCl (0.97 mL, 1.93 mmol) in THF (6 mL). The desired compound was purified by column chromatography (EtOAc/Hex, 1:4 *v*/*v*). Yield: 286 mg (70%), yellow crystalline solid, mp = 78–79 °C, R*_f_* = 0.23 (EtOAc/Hex, 1:5 *v*/*v*). ^1^H NMR (700 MHz, CDCl_3_): δ 0.98 (3H, d, *J* = 6.8 Hz, CH_3_), 1.05 (3H, d, *J* = 6.8 Hz, CH_3_), 2.12 (1H, s, OH), 2.15–2.12 (1H, m, C*H*-(CH_3_)_2_), 4.70 (1H, d, *J* = 6.3 Hz, C*H*-OH), 7.26–7.32 (1H, m, NPh, 4-H), 7.34–7.39 (3H, m, CPh 3,4,5-H), 7.43–7.47 (2H, m, NPh 3,4-H), 7.55–7.59 (2H, m, CPh 2,6-H), 7.67–7.76 (2H, m, NPh 2,6-H), 7.91 (1H, s, 5-H). ^13^C NMR (176 MHz, CDCl_3_): δ 18.1 (CH_3_), 18.8 (CH_3_), 34.9 (*C*H-(CH_3_)_2_), 72.2 (CH-OH), 81.1 (*C*≡CPh), 93.7(C≡*C*Ph), 119.2 (NPh C-2,6), 122.7 (CPh C-1), 125.3 (C-5), 127.0 (NPh C-4), 128.5 (CPh C-3,5), 128.8 (CPh C-4), 129.4 (C-4), 129.6 (NPh C-3,5), 131.8 (CPh C-2,6), 134.9 (C-3), 139.7 (NPh C-1). ^15^N NMR (71 MHz, CDCl_3_): δ −163.8 (N-1), N-2 not found. IR (ν, cm^−1^): 3383 (OH), 3054 (CH_arom_), 2959, 2872 (CH_alif_), 1596, 1501, 1458, 1376, 1331, 1219 (C=C, C=N, C–N), 1056, 1033 (CH-OH), 963, 752, 688 (CH=CH of monosubstituted benzenes). MS (ES^+^): *m*/*z* (%): 317 ([M + H]^+^, 99). HRMS (ESI) for C_21_H_20_N_2_ONa ([M + Na]^+^): requires 339.1468 and found 339.1467.

##### Phenyl[1-phenyl-3-(phenylethynyl)-1*H*-pyrazol-4-yl]methanol **7**

Phenyl[1-phenyl-3-(phenylethynyl)-1*H*-pyrazol-4-yl]methanol **7** was prepared in accordance with general procedure (A) from 1-phenyl-3-(2-phenylethynyl)-1*H*-pyrazole-4-carbaldehyde **2** (100 mg, 0.37 mmol) and PhMgBr (0.15 mL, 0.44 mmol) in DCM (2 mL). The desired compound was purified by column chromatography (EtOAc/Hex, 1:7 *v*/*v*). Yield: 105 mg (81%), colourless liquid, R*_f_* = 0.35 (EtOAc/Hex, 1:3 *v*/*v*). ^1^H NMR (700 MHz, CDCl_3_): δ 2.54 (1H, br s, OH), 6.06 (1H, s, CH), 7.27–7.31 (1H, m, NPh 4-H), 7.31–7.33 (1H, m, C4Ph 4-H), 7.33–7.36 (3H, m, C3-Ph 3,4,5-H), 7.37–7.40 (2H, m, C4Ph 3,5-H), 7.41–7.45 (2H, m, NPh 3,5-H), 7.48–7.51 (2H, m, C3-Ph 2,6-H), 7.52–7.54 (2H, m, C4Ph 2,6-H), 7.65–7.70 (2H, m, NPh 2,6-H), 7.80 (1H, s, 5-H). ^13^C NMR (176 MHz, CDCl_3_): δ 68.8 (CH), 80.5 (*C*≡CPh), 94.0 (C≡*C*Ph), 119.2 (NPh C-2,6), 122.4 (C3-Ph C-1), 125.5 (C-5), 126.4 (C4Ph C-2,6), 126.9 (NPh C-4), 127.9 (C4Ph C-1), 128.3 (C3-Ph C-3,5), 128.5 (C4Ph C-3,5), 128.7 (C3-Ph C-4), 129.4 (NPh C-3,5), 130.2 (C-4), 131.7 (C3-Ph C-2,6), 134.8 (C-3), 139.5 (NPh C-1), 142.6 (C4Ph C-1). ^15^N NMR (71 MHz, CDCl_3_): δ −163.9 (N-1). MS (ES^+^): *m*/*z* (%): 351 ([M + H]^+^, 95). HRMS (ESI) for C_24_H_18_N_2_ONa ([M + Na]^+^): requires 373.1311 and found 373.1311.

#### 3.2.3. General Procedure (B) for the Synthesis of Azide–Alkynes **8**–**12**

To a solution of appropriate pyrazole alcohol **3**–**7** (1 equivalent) in dry DCM, TMSN_3_ (1.5 equivalents) and BF_3_∙Et_2_O (0.2 equivalents) were added dropwise. The reaction mixture was stirred for 10–60 min under an argon atmosphere at room temperature. Upon completion (monitored by TLC), the reaction mixture was diluted with an aqueous NaHCO_3_ solution (10 mL) and extracted with DCM (3 × 25 mL). The combined organic layers were dried over anhydrous Na_2_SO_4_, filtered, and evaporated under reduced pressure. The residue was purified by column chromatography.

##### 4-(Azidomethyl)-1-phenyl-3-(phenylethynyl)-1*H*-pyrazole **8**

4-(Azidomethyl)-1-phenyl-3-(phenylethynyl)-1*H*-pyrazole **8** was prepared in accordance with general procedure (B) from [1-phenyl-3-(2-phenylethynyl)-1*H*-pyrazol-4-yl]methanol **3** (100 mg, 0.36 mmol), TMSN_3_ (0.07 mL, 0.55 mmol), and BF_3_∙Et_2_O (0.01 mL, 0.07 mmol) in DCM (1.5 mL). The desired compound was obtained after purification by column chromatography (EtOAc/Hex, 1:8 *v*/*v*). Yield: 54 mg (50%), light yellow liquid, R*_f_* = 0.71 (EtOAc/Hex, 1:3 *v*/*v*). ^1^H NMR (700 MHz, CDCl_3_): δ 4.44 (2H, s, C*H*_2_N_3_), 7.31–7.35 (1H, m, NPh 4-H), 7.35–7.40 (3H, m, CPh 3,4,5-H), 7.45–7.49 (2H, m, NPh 3,5-H), 7.58–7.63 (2H, m, CPh 2,6-H), 7.69–7.74 (2H, m, NPh 2,6-H), 7.96 (1H, s, 5-H). ^13^C NMR (176 MHz, CDCl_3_): δ 44.8 (*C*H_2_N_3_), 79.9 (*C*≡CPh), 94.1 (C≡*C*Ph), 119.5 (NPh C-2,6), 120.9 (C-4), 122.4 (CPh C-1), 126.7 (C-5), 127.4 (NPh C-4), 128.5 (CPh C-3,5), 129.0 (CPh C-4), 129.7 (NPh C-3,5), 132.0 (CPh C-2,6), 136.5 (C-3), 139.6 (NPh C-1). ^15^N NMR (71 MHz, CDCl_3_): δ −162.2 (N-1), −306.6 and −132.9 (N_3_, one not found), N-2 not found. IR (ν, cm^−1^): 3050 (CH_arom_), 2921 (CH_aliph_), 2087 (N_3_), 1595, 1501, 1331, 1250 (C=C, C=N, C–N), 753, 688 (CH=CH of monosubstituted benzenes). MS (ES^+^): *m*/*z* (%): 300 ([M + H]^+^, 99). HRMS (ESI) for C_18_H_14_N_5_ ([M + H]^+^): requires 300.1242 and found 300.1244; for C_18_H_13_N_5_Na ([M + Na]^+^): requires 322.1065 and found 322.1063.

##### 4-(1-Azidoethyl)-1-phenyl-3-(phenylethynyl)-1*H*-pyrazole **9**

4-(1-Azidoethyl)-1-phenyl-3-(phenylethynyl)-1*H*-pyrazole **9** was prepared in accordance with general procedure (B) from 1-[1-phenyl-3-(phenylethynyl)-1*H*-pyrazol-4-yl]ethan-1-ol **4** (205 mg, 0.71 mmol), TMSN_3_ (0.14 mL, 1.07 mmol), and BF_3_∙Et_2_O (0.02 mL, 0,14 mmol) in DCM (2 mL). The desired compound was obtained after purification by column chromatography (EtOAc/Hex, 1:4 *v*/*v*). Yield: 160 mg (72%), colourless oil, R*_f_* = 0.72 (EtOAc/Hex, 1:3 *v*/*v*). ^1^H NMR (700 MHz, CDCl_3_): δ 1.70 (3H, d, *J* = 7.0 Hz, CH_3_), 4.87 (1H, q, *J* = 7.0 Hz, C*H*N_3_), 7.34–7.36 (1H, m, NPh 4-H), 7.39–7.40 (3H, m, CPh 3,4,5-H), 7.48–7.51 (2H, m, NPh 3,5-H), 7.62–7.63 (2H, m, CPh 2,6-H), 7.74–7.75 (2H, m, NPh 2,6-H), 7.94 (1H, s, 5-H). ^13^C NMR (176 MHz, CDCl_3_): δ 20.7 (CH_3_), 52.7 (*C*HN_3_), 80.2 (*C*≡CPh), 94.0 (C≡*C*Ph), 119.3 (NPh C-2,6), 122.4 (CPh C-1), 124.9 (C-5), 126.7 (C-4), 127.2 (NPh C-4), 128.4 (CPh C-3,5), 128.9 (CPh C-4), 129.5 (NPh C-3-5), 131.8 (CPh C-2,6), 135.1 (C-3), 139.5 (NPh C-1). ^15^N NMR (71 MHz, CDCl_3_): δ −294.3 (N_3_), −163.3 (N-1), −133.9 (N_3_), −73.6 (N-2). IR (KBr, ν, cm^−1^): 3146 (C≡CH), 3055 (CH_arom_), 2985, 2936 (CH_aliph_), 2102 (N_3_), 1597, 1549, 1502, 1216 (C=C, C–N), 820, 756, 688 (CH=CH of monosubstituted benzenes). MS (ES^+^): *m*/*z* (%): 314 ([M + H]^+^, 100). HRMS (ESI) for C_19_H_16_N_5_ ([M + H]^+^): requires 314.1400 and found 314.1395.

##### 4-(1-Azidopropyl)-1-phenyl-3-(phenylethynyl)-1*H*-pyrazole **10**

4-(1-Azidopropyl)-1-phenyl-3-(phenylethynyl)-1*H*-pyrazole **10** was prepared in accordance with general procedure (B) from 1-[1-phenyl-3-(phenylethynyl)-1*H*-pyrazol-4-yl]propan-1-ol **5** (100 mg, 0.33 mmol), TMSN_3_ (0.07 mL, 0.5 mmol), and BF_3_∙Et_2_O (0.01 mL, 0.07 mmol) in DCM (1 mL). The desired compound was obtained after purification by column chromatography (EtOAc:Hex, 1:12 *v*/*v*). Yield: 81 mg (75%), yellowish crystalline solid, mp = 74–75 °C, R*_f_* = 0.73 (EtOAc/Hex, 1:3 *v*/*v*). ^1^H NMR (700 MHz, CDCl_3_): δ 1.06 (3H, t, *J* = 7.3Hz, CH_3_), 1.99 (2H, p, *J* = 7.2 Hz, CH_2_), 4.63 (1H, t, *J* = 7.0 Hz, CH-N_3_), 7.31–7.35 (1H, m, NPh 4-H), 7.36–7.40 (3H, m, CPh 3,4,5-H), 7.44–7.50 (2H, m, NPh 3,5-H), 7.57–7.63 (2H, m, CPh 2,6-H), 7.69–7.77 (2H, m, NPh 2,6-H), 7.91 (1H, s, 5-H). ^13^C NMR (176 MHz, CDCl_3_): δ 10.7 (CH_3_), 28.4 (CH_2_), 58.9 (CH), 80.2 (*C*≡CPh), 93.8 (C≡*C*Ph), 119.2 (NPh C-2,6), 122.3 (CPh C-1), 125.1 (C-5), 125.4 (C-4), 127.1 (NPh C-4), 128.4 (CPh C-3,5), 128.8 (CPh C-4), 129.5 (NPh C-3,5), 131.7 (CPh C-2,6), 135.4 (C-3), 139.4 (NPh C-1). ^15^N NMR (71 MHz, CDCl_3_): δ −163.4 (N-1), −134.5 (CH-N=*N*=N), −116.4 (CH-*N*=N=N), −74.4 (N-2). IR (KBr, ν, cm^−1^): 3147 (CH_arom_), 2967, 2934, 2870 (CH_aliph_), 2092 (N_3_), 1598, 1502, 1328, 1215 (C=C, C=N, C–N), 959, 757, 689 (CH=CH of monosubstituted benzenes). MS (ES^+^): *m*/*z* (%): 328 ([M + H]^+^, 99). HRMS (ESI) for C_20_H_17_N_5_Na ([M + Na]^+^): requires 350.1376 and found 350.1376.

##### 4-(Azido-2-methylpropyl)-1-phenyl-3-(phenylethynyl)-1*H*-pyrazole **11**

4-(Azido-2-methylpropyl)-1-phenyl-3-(phenylethynyl)-1*H*-pyrazole **11** was prepared in accordance with general procedure (B) from 2-methyl-1-[1-phenyl-3-(phenylethynyl)-1*H*-pyrazol-4-yl]propan-1-ol **6** (200 mg, 0.63 mmol), TMSN_3_ (0.1 mL, 0.76 mmol), and BF_3_∙Et_2_O (0.02 mL, 0.13 mmol) in DCM (2 mL). The desired compound was obtained after purification by column chromatography (EtOAc/Hex, 1:10 *v*/*v*). Yield: 177 mg (82%), colourless oil, R*_f_* = 0.68 (EtOAc/Hex, 1:4 *v*/*v*). ^1^H NMR (700 MHz, CDCl_3_): δ 1.00 (3H, d, *J* = 6.8 Hz, CH_3_), 1.06 (3H, d, *J* = 6.7 Hz, CH_3_), 2.18–2.25 (1H, m, C*H*CH_3_)_2_), 4.52 (1H, d, *J* = 7.0 Hz, CH-N_3_), 7.31–7.35 (1H, m, NPh 4-H), 7.37–7.40 (3H, m, CPh 3,4,5-H), 7.45–7.50 (2H, m, NPh 3,5-H), 7.56–7.62 (2H, m, CPh 2,6-H), 7.71–7.79 (2H, m, NPh 2,6-H), 7.91 (1H, s, 5-H). ^13^C NMR (176 MHz, CDCl_3_): δ 19.0 (CH_3_), 19.4 (CH_3_), 33.7 (*C*H- (CH_3_)_2_), 64.2 (*C*H-N_3_), 80.5(*C*≡CPh), 93.9 (C≡*C*Ph), 119.4 (NPh C-2,6), 122.5 (CPh C-1), 124.5 (C-4), 125.6 (C-5), 127.3 (NPh C-4), 128.5 (CPh C-3,5), 129.0 (CPh C-4), 129.6 (NPh C-3,5), 131.9 (CPh C-2,6), 135.9 (C-3), 139.6 (NPh C-1). ^15^N NMR (71 MHz, CDCl_3_): δ −299.2 (N_3_), −163.1 (N-1), −134.1 (N_3_). IR (ν, cm^−1^): 3060 (CH_arom_), 2963, 2873 (CH_aliph_), 2093 (N_3_), 1598, 1502, 1329, 1244 (C=C, C=N, C–N), 753, 687 (CH=CH of monosubstituted benzenes). MS (ES^+^): *m*/*z* (%): ([M + H]^+^, 99). HRMS (ESI) for C_21_H_20_N_5_Na ([M + Na]^+^): requires 364.1533 and found 364.1533.

##### 4-[Azido(phenyl)methyl]-1-phenyl-3-(phenylethynyl)-1*H*-pyrazole **12**

4-[Azido(phenyl)methyl]-1-phenyl-3-(phenylethynyl)-1*H*-pyrazole **12** was prepared in accordance with general procedure (B) from phenyl[1-phenyl-3-(phenylethynyl)-1*H*-pyrazol-4-yl]methanol **7** (105 mg, 0.3 mmol), TMSN_3_ (0.16 mL, 0.45 mmol), and BF_3_∙Et_2_O (0.01 mL, 0.06 mmol) in DCM (2 mL). The desired compound was obtained after purification by column chromatography (EtOAc/Hex, 1:7 *v*/*v*). Yield: 105 mg (93%), white crystalline solid, mp = 86–87 °C, R*_f_* = 0.73 (EtOAc/Hex, 1:3 *v*/*v*). ^1^H NMR (700 MHz, CDCl_3_): δ 5.90 (1H, s, CH), 7.30–7.33 (1H, m, NPh 4-H), 7.33–7.39 (4H, m, C4Ph 4-H and C3-Ph 3,4,5-H), 7.40–7.44 (2H, m, C4Ph 3,5-H), 7.44–7.48 (4H, m, C4Ph 2,6-H, NPh 3,5-H), 7.48–7.52 (2H, m, C3-Ph 2,6-H), 7.68–7.72 (2H, m, NPh 2,6-H), 7.82 (1H, s, 5-H). ^13^C NMR (176 MHz, CDCl_3_): δ 60.6 (CH), 80.0 (*C*≡CPh), 94.3 (C≡*C*Ph), 119.2 (NPh C-2,6), 122.3 (C3-Ph C-1), 125.97 (C-5), 126.0 (C-4), 127.1 (NPh C-4), 127.2 (C4Ph C-2,6), 128.3 (C3-Ph C-3,5), 128.4 (C4Ph C-4), 128.78 (C3-Ph C-4), 128.79 (C4Ph C-3,5), 129.4 (NPh C-3,5), 131.7 (C3-Ph C-2,6), 135.4 (C-3), 138.5 (C4Ph C-1), 139.4 (NPh C-1). ^15^N NMR (71 MHz, CDCl_3_): δ −163.4 (N-1), −134.7 (N_3_). IR (KBr, ν, cm^−1^): 3058 (CH_arom_), 2097 (N_3_), 1597, 1502, 1303, 1227 (C=C, C=N, C–N), 958, 751, 686 (CH=CH of monosubstituted benzenes). MS (ES^+^): *m*/*z* (%): 376 ([M + H]^+^, 99). HRMS (ESI) for C_24_H_17_N_5_Na ([M + Na]^+^): requires 398.1376 and found 398.1376.

#### 3.2.4. General Procedure (C) for the Synthesis of 7-Iodo-2*H*-pyrazolo[4,3-*c*]pyridines **13**–**17**

To a solution of appropriate azide–alkyne **8–12** (1 equivalent) in DCM, the appropriate base K_3_PO_4_ (5 equivalents) or NaHCO_3_ (1 equivalent) and I_2_ (5 equivalents) were added. The reaction mixture was stirred at room temperature for 12 h. Upon completion (monitored by TLC), the reaction mixture was diluted with an aqueous Na_2_S_2_O_4_ solution (20 mL) and extracted with EtOAc (3 × 25 mL). The combined organic layers were dried over anhydrous Na_2_SO_4_, filtered, and evaporated under reduced pressure. The residue was purified by column chromatography.

##### 7-Iodo-2,6-diphenyl-2*H*-pyrazolo[4,3-*c*]pyridine **13**

7-Iodo-2,6-diphenyl-2*H*-pyrazolo[4,3-*c*]pyridine **13** was prepared in accordance with general procedure (C) from 4-(azidomethyl)-1-phenyl-3-(2-phenylethynyl)-1*H*-pyrazole **8** (276 mg, 0.92 mmol), K_3_PO_4_ (978 mg, 4.6 mmol), and I_2_ (1.472 g, 4.6 mmol) in DCM (9.8 mL). The desired compound was obtained after purification by column chromatography (EtOAc/Hex, 1:2 *v*/*v*). Yield: 299 mg (82%), light yellow crystalline solid, mp = 110–111 °C, R*_f_* = 0.13 (EtOAc/Hex, 1:3 *v*/*v*). ^1^H NMR (700 MHz, CDCl_3_): δ 7.42–7.45 (1H, m, CPh 4-H), 7.46–7.52 (3H, m, CPh 3,5-H and NPh 4-H), 7.55–7.59 (2H, m, NPh 3,5-H), 7.68–7.74 (2H, m, CPh 2,6-H), 7.93–7.99 (2H, m, NPh 2,6-H), 8.77 (1H, s, 3-H), 9.13 (1H, s, 4-H). ^13^C NMR (176 MHz, CDCl_3_): δ 82.1 (C-7), 118.5 (C-3a), 121.8 (NPh, C-2,6), 123.3 (C-3), 128.1 (CPh, C-3,5), 128.4 (CPh, C-4), 129.3 (NPh, C-4), 129.9 (NPh, C-3,5), 130.1 (CPh, C-2,6), 139.9 (NPh, C-1), 142.2 (CPh, C-1), 146.1 (C-4), 153.5 (C-7a), 155.7 (C-6). ^15^N NMR (71 MHz, CDCl_3_): δ −146.1 (N-2), −90.6 (N-1), −78.4 (N-5). IR (ν, cm^−1^): 3044, 3035 (CH_arom_), 1604, 1590, 1505, 1465, 1202 (C=C, C=N, C–N), 741, 700, 679 (CH=CH of monosubstituted benzenes). MS (ES^+^): *m*/*z* (%): 398 ([M + H]^+^, 100). HRMS (ESI) for C_18_H_13_N_3_I ([M + H]^+^): requires 398.0149 and found 398.0149.

##### 7-Iodo-4-methyl-2,6-diphenyl-2*H*-pyrazolo[4,3-*c*]pyridine **14**

7-Iodo-4-methyl-2,6-diphenyl-2*H*-pyrazolo[4,3-*c*]pyridine **14** was prepared in accordance with general procedure (C) from 4-(1-azidoethyl)-1-phenyl-3-(phenylethynyl)-1*H*-pyrazole **9** (255 mg, 0.79 mmol), NaHCO_3_ (69 mg, 0.82 mmol), and I_2_ (1034 mg, 4.07 mmol) in DCM (8.1 mL). The desired compound was obtained after purification by column chromatography (EtOAc/Hex, 1:2 *v*/*v*). Yield: 238 mg (72%), light yellow crystalline solid, mp = 186–189 °C, R*_f_* = 0.43 (EtOAc/Hex, 1:2 *v*/*v*). ^1^H NMR (700 MHz, CDCl_3_): δ 2.83 (s, 3H, CH_3_), 7.40–7.43 (m, 1H, CPh 4-H), 7.46–7.49 (m, 3H, CPh 3,5-H; NPh 4-H), 7.55–7.57 (m, 2H, NPh 3,5-H), 7.67–7.69 (m, 2H, CPh 2,6-H), 7.95–7.96 (m, 2H, NPh 2,6-H), 8.72 (s, 1H, 3-H). ^13^C NMR (176 MHz, CDCl_3_): δ 22.5 (CH_3_), 79.0 (C-7), 119.0 (C-3a), 121.6 (NPh C-2,6) 123.0 (C-3), 128.0 (CPh C-3,5), 128.3 (CPh C-4), 129.0 (NPh C-4), 129.8 (NPh C-3,5), 130.0 (CPh C-2,6), 140.0 (NPh C-1), 142.5 (CPh C-1), 153.5 (C-7a), 155.3 (C-4), 155.6 (C-6). ^15^N NMR (71 MHz, CDCl_3_): δ −147.5 (N-2), −88.5 (N-1), −80.5 (N-5). IR (KBr, ν, cm^−1^): 3131, 3107 (CH_arom_), 2956 (CH_aliph_), 1586, 1504, 1370, 1205 (C=C, C=N, C–N), 798, 768, 750, 696 (CH=CH of monosubstituted benzenes). MS (ES^+^): *m/z* (%): 412 ([M + H]^+^, 100). HRMS (ESI) C_19_H_15_IN_3_ ([M + H]^+^): requires 412.0305 and found 412.0304.

##### 4-Ethyl-7-iodo-2,6-diphenyl-2*H*-pyrazolo[4,3-*c*]pyridine **15**

4-Ethyl-7-iodo-2,6-diphenyl-2*H*-pyrazolo[4,3-*c*]pyridine **15** was prepared in accordance with general procedure (C) from 4-(1-azidopropyl)-1-phenyl-3-(phenylethynyl)-1*H*-pyrazole **10** (329 mg, 1.01 mmol), NaHCO_3_ (85 mg, 1.01 mmol), and I_2_ (1278 mg, 5.03 mmol) in DCM (10 mL) the desired compound was obtained after purification by column chromatography (EtOAc/Hex, 1:6 *v*/*v*). Yield: 348 mg (81%), orange crystalline solid, mp = 145–146 °C, R*_f_* = 0.39 (EtOAc/Hex, 1:3 *v*/*v*). ^1^H NMR (700 MHz, CDCl_3_): δ 1.46 (3H, t, *J* = 7.6 Hz, CH_3_), 3.14 (2H, q, *J* = 7.6 Hz, CH_2_), 7.40–7.43 (1H, m, CPh 4-H), 7.45–7.50 (3H, m, CPh 3,5-H; NPh 4-H), 7.54–7.58 (2H, m, NPh 3,5-H), 7.69–7.75 (2H, m, CPh 2,6-H), 7.93–7.98 (2H, m, NPh 2,6-H), 8.74 (1H, s, 3-H). ^13^C NMR (176 MHz, CDCl_3_): δ 13.5 (CH_3_), 29.7 (CH_2_), 78.8 (C-7), 117.8 (C-3a), 121.5 (NPh C-2,6), 122.6 (C-3), 127.8 (CPh C-3,5), 128.1 (CPh C-4), 128.9 (NPh C-4), 129.7 (NPh C-3,5), 130.0 (CPh, C-2,6), 139.8 (NPh C-1), 142.4 (CPh C-1), 153.8 (C-7a), 155.2 (C-6), 160.2 (C-4). ^15^N NMR (71 MHz, CDCl_3_): δ −148.2 (N-2), −89.9 (N-1), −82.2 (N-5). IR (KBr, ν, cm^−1^): 3059 (CH_arom_), 2969, 2929 (CH_aliph_), 1584, 1504, 1464, 1374, 1273, 1201 (C=C, C=N, C–N), 905, 768, 698 (CH=CH of monosubstituted benzenes). MS (ES^+^): *m/z* (%): 425 ([M + H]^+^, 99). HRMS (ESI) C_20_H_17_IN_3_ ([M + H]^+^): requires 426.0462 and found 426.0462.

##### 7-Iodo-4-isopropyl-2,6-diphenyl-2*H*-pyrazolo[4,3-*c*]pyridine **16**

7-Iodo-4-isopropyl-2,6-diphenyl-2*H*-pyrazolo[4,3-*c*]pyridine **16** was prepared in accordance with general procedure (C) from 4-(azido-2-methylpropyl)-1-phenyl-3-(phenylethynyl)-1*H*-pyrazole **11** (177 mg, 0.52 mmol), NaHCO_3_ (44 mg, 0.52 mmol), and I_2_ (659 mg, 2.6 mmol) in DCM (5.2 mL). The desired compound was obtained after purification by column chromatography (EtOAc/Hex, 1:6 *v*/*v*). Yield: 191 mg (84%), white crystalline solid, mp = 134–135 °C, R*_f_* = 0.50 (EtOAc/Hex, 1:5 *v*/*v*). ^1^H NMR (700 MHz, CDCl_3_): δ 1.49 (6H, d, *J* = 7.0 Hz, CH-(C*H*_3_)_2_), 3.47 (1H, hept, *J* = 7.0 Hz, C*H*-(CH_3_)_2_), 7.39–7.44 (1H, m, CPh 4-H), 7.45–7.51 (3H, m, CPh 3,5-H and NPh 4-H), 7.54–7.59 (2H, m, NPh 3,5-H), 7.74–7.82 (2H, m, CPh 2,6-H), 7.93–7.99 (2H, m, NPh 2,6-H), 8.76 (1H, s, 3-H). ^13^C NMR (176 MHz, CDCl_3_): δ 22.1 (CH-(*C*H_3_)_2_), 35.6 (C*H*-(CH_3_)_2_), 78.8 (C-7), 117.0 (C-3a), 121.7 (NPh C-2,6), 122.5 (C-3), 127.8 (CPh C-3,5), 128.2 (CPh C-4), 129.0 (NPh C-4), 129.8 (NPh C-3,5), 130.4 (CPh C-2,6), 140.1 (NPh C-1), 142.5 (CPh C-1), 154.3 (C-7a), 154.9 (C-6), 163.8 (C-4). ^15^N NMR (71 MHz, CDCl_3_): δ −158.9 (N-2), −90.3 (N-1), −82.7 (N-5). IR (ν, cm^−1^): 3114, 3083, 3062 (CH_arom_), 2970, 2928, 2868 (CH_aliph_), 1584, 1507, 1468, 1391, 1212 (C=C, C=N, C–N), 1106, 1023, 915, 763, 697 (CH=CH of monosubstituted benzenes). MS (ES^+^): *m/z* (%): 440 ([M + H]^+^, 97.7). HRMS (ESI) for C_21_H_19_N_3_I ([M + H]^+^): requires 440.0618 and found 440.0618.

##### 7-Iodo-2,4,6-triphenyl-2*H*-pyrazolo[4,3-*c*]pyridine **17**

7-Iodo-2,4,6-triphenyl-2*H*-pyrazolo[4,3-*c*]pyridine **17** was prepared in accordance with general procedure (C) from 4-[azido(phenyl)methyl]-1-phenyl-3-(phenylethynyl)-1*H*-pyrazole **12** (62 mg, 0.165 mmol), K_3_PO_4_ (210 mg, 0.83 mmol), and I_2_ (175 mg, 0.83 mmol) in DCM (1.7 mL). The desired compound was obtained after purification by column chromatography (EtOAc/Hex, 1:8 *v*/*v*). Yield: 69 mg (88%), white crystalline solid, mp = 93–94 °C, R*_f_* = 0.59 (EtOAc/Hex, 1:3 *v*/*v*). ^1^H NMR (700 MHz, CDCl_3_): δ 7.43–7.49 (2H, m, C6Ph 4-H and NPh 4-H), 7.49–7.58 (7H, m, C6Ph, NPh and C4Ph 3,5-H; C4Ph 4-H), 7.80–7.87 (2H, m, C6Ph 2,6-H), 7.94–8.00 (2H, m, NPh 2,6-H), 8.04–8.12 (2H, m, C4Ph 2,6-H), 8.90 (1H, s, 3-H). ^13^C NMR (176 MHz, CDCl_3_): δ 80.3 (C-7), 116.7 (C-3a), 121.5 (NPh C-2,6), 123.7 (C-3), 127.8 (C6Ph C-3,5), 128.2 (C6Ph C-4), 128.4 (C4Ph C-2,6), 128.9 (C4Ph C-3,5), 129.0 (NPh C-4), 129.7 (NPh C-3,5), 129.9 (C4Ph C-4), 130.2 (C6Ph C-2,6), 138.4 (C4Ph C-1), 139.7 (NPh C-1), 142.3 C6Ph C-1), 154.5 (C-7a), 154.7 (C-4), 155.4 (C-6). ^15^N NMR (71 MHz, CDCl_3_): δ −146.4 (N-2), −90.3 (N-1), N-5 not found. IR (KBr, ν, cm^−1^): 3058 (CH_arom_), 2922 (CH_aliph_), 1570, 1505, 1464, 1371, 1212 (C=C, C=N, C–N), 969, 750, 698 (CH=CH of monosubstituted benzenes). MS (ES^+^): *m*/*z* (%): 474 ([M + H]^+^, 99). HRMS (ESI) C_24_H_17_IN_3_ ([M + H]^+^): requires 474.0462 and found 474.0462.

#### 3.2.5. General Procedure (D) for the Synthesis of 7-Substituted Pyrazolo[4,3-*c*]pyridine derivatives **18**–**39** by Suzuki–Miyaura Cross-Coupling with Boronic acids

To a solution of appropriate7-iodo-2*H*-pyrazolo[4,3-*c*]pyridine **13−17** (1 equivalent) in a mixture of EtOH and water (3:1, *v*/*v*), boronic acid (1.2 equivalents), Cs_2_CO_3_ (2 equivalents), and Pd(OAc)_2_ (0.07 equivalents) were added under argon atmosphere. The mixture was stirred at 100 °C under microwave irradiation (100 W and 300 Pa) for 0.5–1 h. Upon completion (monitored by TLC), the reaction mixture was cooled to room temperature and filtered through a pad of Celite, and the filter cake was washed with EtOAc (20 mL). The filtrate was diluted with water (20 mL) and extracted with EtOAc (3 × 25 mL). The combined organic layers were washed with brine (10 mL), dried over anhydrous Na_2_SO_4_, filtered, and evaporated under reduced pressure. The residue was purified by column chromatography.

##### 2,6,7-Triphenyl-2*H*-pyrazolo[4,3-*c*]pyridine **18**

2,6,7-Triphenyl-2*H*-pyrazolo[4,3-*c*]pyridine **18** was prepared in accordance with general procedure (D) from 7-iodo-2,6-diphenyl-2*H*-pyrazolo[4,3-*c*]pyridine **13** (60 mg, 0.15 mmol), phenylboronic acid (22 mg, 0.18 mmol), Cs_2_CO_3_ (98 mg, 0.3 mmol), Pd(OAc)_2_ (2.4 mg, 0.01 mmol), EtOH (0.9 mL), and water (0.3 mL). The reaction was finished after 30 min. The desired compound was obtained after purification by column chromatography (EtOAc/Hex, 1:4 to 1:2 *v*/*v*). Yield: 50 mg (96%), brown crystalline solid, mp = 151–152 °C, R*_f_* = 0.08 (EtOAc/Hex, 1:3 *v*/*v*). ^1^H NMR (700 MHz, CDCl_3_): δ 7.21–7.26 (3H, m, C6Ph 3,4,5-H), 7.28–7.31 (1H, m, C7Ph 4-H), 7.31–7.35 (2H, m, C7Ph 3,5-H), 7.42–7.46 (3H, m, C6Ph 2,6-H and NPh 4-H), 7.49–7.55 (4H, m, NPh, 3,5-H and C7Ph 2,6-H), 7.89–7.94 (2H, m NPh 2,6-H), 8.66 (1H, s, 3-H), 9.32 (1H, s, 4-H). ^13^C NMR (176 MHz, CDCl_3_): δ 120 (C-3a), 121.4 (NPh C-2.6), 121.8 (C-3), 123.1 (C-6), 127.2 (C6Ph C-4), 127.3 (C7Ph C-4), 127.8 (C6Ph C-2,6), 128.0 (C7Ph C-3,5), 128.7 (NPh C-4), 129.6 (NPh C-3,5), 130.5 (C7Ph C-2,6), 131.1 (C7Ph C-2,6), 135.6 (C7Ph C-1), 140.0 (NPh C-1), 140.6 (C6Ph C-1), 145.7 (C-4), 149.3 (C-4), 151.2 (C-7a). ^15^N NMR (71 MHz, CDCl_3_): δ −144.9 (N-2), −96.8 (N-1), −80.5 (N-5). IR (ν, cm^−1^): 3502, 3081 (CH_arom_), 1663, 1610, 1592, 1504, 1203, 1180, 1127 (C=C, C=N, C–N), 761, 697, 688 (CH=CH of monosubstituted benzenes). MS (ES^+^): *m/z* (%): 348 ([M + H]^+^, 98). HRMS (ESI) for C_24_H_18_N_3_ ([M + H]^+^): requires 348.1495 and found 348.1495.

##### 7-(2-Methoxyphenyl)-2,6-diphenyl-2*H*-pyrazolo[4,3-*c*]pyridine **19**

7-(2-Methoxyphenyl)-2,6-diphenyl-2*H*-pyrazolo[4,3-*c*]pyridine **19** was prepared in accordance with general procedure (D) from 7-iodo-2,6-diphenyl-2*H*-pyrazolo[4,3-*c*]pyridine **13** (60 mg, 0.15 mmol), (2-methoxyphenyl)boronic acid (27 mg, 0.18 mmol), Cs_2_CO_3_ (98 mg, 0.3 mmol), Pd(OAc)_2_ (2.4 mg, 0.01 mmol), EtOH (0.9 mL), and water (0.3 mL). The reaction was stopped after 1 h. The desired compound was obtained after purification by column chromatography (EtOAc/Hex, 1:3 *v*/*v*). Yield: 23 mg (40%), light yellow crystalline solid, mp = 169–170 °C, R*_f_* = 0.08 (EtOAc/Hex, 1:3 *v*/*v*). ^1^H NMR (700 MHz, CDCl_3_): δ 3.42 (3H, s, OCH_3_), 6.82–6.87 (1H, m, C7Ph), 7.00–7.03 (1H, m, C7Ph), 7.17–7.21 (1H, m, C7Ph), 7.21–7.24 (2H, m, Ph), 7.30–7.35 (1H, m, Ph,), 7.39–7.42 (1H, m, Ph), 7.43–7.46 (2H, m, Ph), 7.47–7.51 (3H, m, Ph), 7.81–7.92 (2H, m, NPh, 2,6-H), 8.61 (s, 1H, 3-H), 9.34 (s, 1H, 4-H). ^13^C NMR (176 MHz, CDCl_3_): δ 55.3, 111.6, 119.7, 120.1, 120.7, 121.6, 121.8, 125.1, 127.1, 127.6, 128.7, 129.3, 129.6, 129.7, 132.5, 140.3, 141.5, 145.9, 150.4, 151.6, 157.0. IR (ν, cm^−1^): 3062, 3019 (CH_arom_), 2922, 2852 (CH_aliph_), 1600, 1590, 1501, 1478, 1435, 1242, 1233, 1203 (C=C, C=N, C–N), 1112, 1043, 1021 (C-O-C), 763, 750, 697, 686 (CH=CH of mono- and disubstituted benzenes). MS (ES^+^): *m*/*z* (%): 378 ([M + H]^+^, 99). HRMS (ESI) for C_25_H_20_N_3_O ([M + H]^+^): requires 378.1601 and found 378.1601.

##### 7-(3-Methoxyphenyl)-2,6-diphenyl-2*H*-pyrazolo[4,3-*c*]pyridine **20**

7-(3-Methoxyphenyl)-2,6-diphenyl-2*H*-pyrazolo[4,3-*c*]pyridine **20** was prepared in accordance with general procedure (D) from 7-iodo-2,6-diphenyl-2*H*-pyrazolo[4,3-*c*]pyridine **13** (60 mg, 0.15 mmol), (3-methoxyphenyl)boronic acid (27 mg, 0.18 mmol), Cs_2_CO_3_ (98 mg, 0.3 mmol), Pd(OAc)_2_ (2.4 mg, 0.01 mmol), EtOH (0.9 mL), and water (0.3 mL). The reaction was finished after 1 h. The desired compound was obtained after purification by column chromatography (EtOAc/Hex, 1:3 *v*/*v*). Yield: 44 mg (78%), white crystalline solid, mp = 71–72°C, R*_f_* = 0.08 (EtOAc/Hex, 1:3 *v*/*v*). ^1^H NMR (700 MHz, CDCl_3_): δ 3.66 (3H, s, OCH_3_), 6.81–6.87 (1H, m, C7Ph 4-H), 7.02–7.08 (1H, m, C7Ph 2-H), 7.10–7.14 (1H, m, C7Ph 6-H), 7.21–7.29 (4H, m, C7Ph 5-H, C6Ph 3,4,5-C), 7.40–7.44 (1H, m, NPh 4-H), 7.44–7.48 (2H, m, C6Ph 2,6-H), 7.49–7.54 (2H, m, NPh 3,5-H), 7.86–7.94 (2H, m, NPh 2,6-H), 8.64 (1H, s, 3-H), 9.30 (1H, s, 4-H). ^13^C NMR (176 MHz, CDCl_3_): δ 55.3 (OCH_3_), 113.7 (C7Ph C-4), 116.6 (C7Ph C-2), 120.2 (C-3a), 121.5 (NPh C-2,6), 121.9 (C-3), 123.0 (C-7), 123.9 (C7Ph C-6), 127.3 (C6Ph C-4), 128.0 (C6Ph C-3,5), 128.8 (NPh C-4), 129.1 (C7Ph C-5), 129.8 (NPh C-3,5), 130.6 (C6Ph C-2,6), 137.0 (C7Ph C-1), 140.2 (NPh C-1), 140.9 (C6Ph C-1), 145.9 (C-4), 149.6 (C-6), 151.2 (C-7a), 159.3 (C7Ph C-3). ^15^N NMR (71 MHz, CDCl_3_): δ −145.2 (N-2), −96.9 (N-1), −79.4 (N-5). IR (ν, cm^−1^): 3394, 3058, 3011 (CH_arom_), 2920, 2849 (CH_aliph_), 1592, 1575, 1507, 1464, 1367, 1317, 1286, 1212 (C=C, C=N, C–N), 1150, 1051 (C-O-C), 764, 756, 699, 689 (CH=CH of mono- and disubstituted benzenes). MS (ES^+^): *m/z* (%): 378 ([M + H]^+^, 99). HRMS (ESI) for C_25_H_20_N_3_O ([M + H]^+^): requires 378.1601 and found 378.1601.

##### 7-(4-Methoxyphenyl)-2,6-diphenyl-2*H*-pyrazolo[4,3-*c*]pyridine **21**

7-(4-Methoxyphenyl)-2,6-diphenyl-2*H*-pyrazolo[4,3-*c*]pyridine **21** was prepared in accordance with general procedure (D) from 7-iodo-2,6-diphenyl-2*H*-pyrazolo[4,3-*c*]pyridine **13** (60 mg, 0.15 mmol), (4-methoxyphenyl)boronic acid (27 mg, 0.18 mmol), Cs_2_CO_3_ (98 mg, 0.3 mmol), Pd(OAc)_2_ (2.4 mg, 0.01 mmol), EtOH (0.9 mL), and water (0.3 mL). The reaction was finished after 1 h. The desired compound was obtained after purification by column chromatography (EtOAc/Hex, 1:3 *v*/*v*). Yield: 44 mg (78%), white crystalline solid, mp = 192–193 °C, R*_f_* = 0.08 (EtOAc/Hex, 1:3 *v*/*v*). ^1^H NMR (700 MHz, CDCl_3_): δ 3.82 (3H, s, OCH_3_), 6.82–6.93 (2H, m, C7Ph 3,5-H), 7.21–7.25 (1H, m, C6Ph 4-H), 7.25–7.30 (2H, m, C6Ph 3,5-H), 7.41–7.49 (5H, m, NPh 4-H, C6Ph 2,6-H, C7Ph 2,6-H), 7.50–7.56 (2H, m, NPh 3,5-H), 7.84–7.97 (2H, m, NPh 2,6-H), 8.63 (1H, s, 3-H), 9.28 (1H, s, 4-H). ^13^C NMR (176 MHz, CDCl_3_): δ 55.2 (OCH_3_), 113.5 (C7Ph C-3,5), 120.1 (C-3a), 121.4 (NPh C-2,6), 121.7 (C-3), 122.6 (C-7), 127.0 (C6Ph C-4), 127.8 (C7Ph C-1), 127.9 (C6Ph C-3,5), 128.7 (NPh C-4), 129.6 (NPh C-3,5), 130.5 (C6Ph C-2,6), 132.4 (C7Ph C-2,6), 140.1 (NPh C-1), 140.9 (C6Ph C-1), 145.3 (C-4), 149.2 (C-6), 151.4 (C-7a), 158.9 (C7Ph C-4). ^15^N NMR (71 MHz, CDCl_3_): δ −145.4 (N-2), −97.0 (N-1), −79.1 (N-5). IR (ν, cm^−1^): 3135, 3062, 3020 (CH_arom_), 2927, 2837 (CH_aliph_), 1589, 1503, 1438, 1290, 1252 (C=C, C=N, C–N), 1178, 1031 (C-O-C), 763, 758, 689 (CH=CH of mono- and disubstituted benzenes). MS (ES^+^): *m/z* (%): 378 ([M + H]^+^, 100). HRMS (ESI) for C_25_H_20_N_3_O ([M + H]^+^): requires 378.1603 and found 378.1601.

##### 7-(3,4-Dimethoxyphenyl)-2,6-diphenyl-2*H*-pyrazolo[4,3-*c*]pyridine **22**

7-(3,4-Dimethoxyphenyl)-2,6-diphenyl-2*H*-pyrazolo[4,3-*c*]pyridine **22** was prepared in accordance with general procedure (D) from 7-iodo-2,6-diphenyl-2*H*-pyrazolo[4,3-*c*]pyridine **13** (60 mg, 0.15 mmol), (3,4-dimethoxyphenyl)boronic acid (33 mg, 0.18 mmol), Cs_2_CO_3_ (98 mg, 0.3 mmol), Pd(OAc)_2_ (2.4 mg, 0.01 mmol), EtOH (0.9 mL), and water (0.3 mL). The reaction was finished after 1 h. The desired compound was obtained after purification by column chromatography (EtOAc/Hex, 1:3 *v*/*v*). Yield: 44 mg (72%), white crystalline solid, mp = 163–164 °C, R*_f_* = 0.08 (EtOAc/Hex, 1:3 *v*/*v*). ^1^H NMR (700 MHz, CDCl_3_): δ 3.61 (3H, s, 3-OCH_3_), 3.90 (3H, s, 4-OCH_3_), 6.85–6.90 (1H, m, C7Ph 5-H), 6.91–6.95 (1H, m, C7Ph 2-H), 7.21–7.30 (4H, m, C6Ph 3,4,5-H, C7Ph 6-H), 7.42–7.45 (1H, m, NPh 4H), 7.45–7.49 (2H, m, NPh 2,6-H), 7.50–7.56 (2H, m, NPh 3,5-H), 7.87–7.98 (2H, m, NPh 2,6-H), 8.65 (1H, s, 3-H), 9.28 (1H, s, 4-H). ^13^C NMR (176 MHz, CDCl_3_): δ 55.7 (3-OCH_3_), 55.9 (4-OCH_3_), 110.9 (C7Ph C-5), 114.9 (C7Ph C-2), 120.3 (C-3a), 121.5 (NPh C-2,6), 121.9 (C-3), 122.8 (C-7), 124.0 (C7Ph C-6), 127.2 (C6Ph C-4), 128.0 (C7Ph C-1), 128.1 (C6Ph C-3,5), 128.8 (NPh C-4), 129.8 (NPh C-3,5), 130.5 (C6Ph C-2,6), 140.2 (NPh C-1), 141.2 (C6Ph C-1), 145.4 (C-4), 148.4 (C7Ph C-3,4), 149.4 (C-6), 151.4 (C-7a). ^15^N NMR (71 MHz, CDCl_3_): δ −145.4 (N-2), −97.1 (N-1), −79.2 (N-5). IR (ν, cm^−1^): 3042, 3019 (CH_arom_), 2967, 2919, 2850 (CH_aliph_), 1592, 1507, 1468, 1253, 1225 (C=C, C=N, C–N), 1141, 1014 (C-O-C), 753, 699, 688 (CH=CH of mono- and disubstituted benzenes). MS (ES^+^): *m/z* (%): 408 ([M + H]^+^, 100). HRMS (ESI) for C_26_H_22_N_3_O_2_ ([M + H]^+^): requires 408.1707 and found 408.1707.

##### 4-(2,6-Diphenyl-2*H*-pyrazolo[4,3-*c*]pyridin-7-yl)phenol **23**

4-(2,6-Diphenyl-2*H*-pyrazolo[4,3-*c*]pyridin-7-yl)phenol **23** was prepared in accordance with general procedure (D) from 7-iodo-2,6-diphenyl-2*H*-pyrazolo[4,3-*c*]pyridine **13** (80 mg, 0.2 mmol), (4-hydroxyphenyl)boronic acid (33 mg, 0.24 mmol), Cs_2_CO_3_ (131 mg, 0.4 mmol), Pd(OAc)_2_ (3 mg, 0.014 mmol), EtOH (1.2 mL), and water (0.4 mL). The reaction was finished after 30 min. The desired compound was obtained after purification by column chromatography (EtOAc/Hex, 1:2 to 2:1 *v*/*v*). Yield: 44 mg (60%), yellowish crystalline solid, mp = 307–308 °C, R*_f_* = 0.18 (EtOAc/Hex, 1:2 *v*/*v*). ^1^H NMR (700 MHz, DMSO-*d*_6_): δ 6.70–6.75 (2H, m, C7Ph 3,5-H), 7.20–7.28 (5H, m, C7Ph 2,6-H; C6Ph 3,4,5-H), 7.37–7.40 (2H, m, C6Ph 2,6-H), 7.49–7.52 (1H, m, NPh 4-H), 7.59–7.63 (2H, m, NPh 3,5-H), 8.04–8.09 (2H, m, NPh 2,6-H), 9.29 (1H, s, 4-H), 9.46 (1H, s, 3-H), 9.50 (1H, s, OH). ^13^C NMR (176 MHz, DMSO-*d*_6_): δ 115.0 (C7Ph C-3,5), 119.7 (C-3a), 121.0 (NPh C-2,6), 122.3 (C-7), 124.2 (C-3), 125.8 (C7Ph C-1), 127.0 C6Ph C-4), 127.6 (C6Ph C-3,5), 128.7 (NPh C-4), 129.8 (NPh C-3,5), 130.3 (C6Ph C-2,6), 132.0 (C7Ph C-2,6), 139.5 (NPh C-1), 140.6 (C6Ph C-1), 145.5 (C-4), 147.8 (C-6), 150.5 C-7a), 156.7 (C7Ph C-4). ^15^N NMR (71 MHz, DMSO-*d*_6_): δ −144.8 (N-2), N-1 and N-5 not found. IR (KBr, ν, cm^−1^): 3449 (OH), 2924 (CH_arom_), 1607, 1592, 1505, 1440, 1273, 1172 (C=C, C=N, C–N), 767, 695 (CH=CH of mono- and disubstituted benzenes). MS (ES^+^): *m/z* (%): 364 ([M + H]^+^, 96). HRMS (ESI) for C_24_H_18_N_3_O ([M + H]^+^): requires 364.1444 and found 364.1446.

##### 4-Methyl-2,6,7-triphenyl-2*H*-pyrazolo[4,3-*c*]pyridine **24**

4-Methyl-2,6,7-triphenyl-2*H*-pyrazolo[4,3-*c*]pyridine **24** was prepared in accordance with general procedure (D) from 7-iodo-4-methyl 2,6-diphenyl-2*H*-pyrazolo[4,3-*c*]pyridine **14** (50 mg, 0.12 mmol), phenylboronic acid (18 mg, 0.145 mmol), Cs_2_CO_3_ (79 mg, 0.24 mmol), Pd(OAc)_2_ (1.9 mg, 0.008 mmol), EtOH (0.9 mL), and water (0.3 mL). The reaction was finished after 30 min. The desired compound was obtained after purification by column chromatography (EtOAc/Hex, 1:4 to 1:2 *v*/*v*). Yield: 116 mg (94%), mp = 202–205 °C, R*_f_* = 0.17 (EtOAc/Hex, 1:3 *v*/*v*). ^1^H NMR (700 MHz, CDCl_3_): δ 2.93 (3H, s, CH_3_), 7.19–7.24 (3H, m, C7Ph), 7.25–7.29 (1H, m, C6Ph 4-H), 7.29–7.33 (2H, m, C6Ph 3,5-H), 7.41–7.48 (5H, m, C6Ph 2,6-H, NPh 4-H and C7Ph), 7.49–7.54 (2H, m, NPh 3,5-H), 7.89–7.91 (2H, m, NPh 2,6-H), 8.61 (1H, s, 3-H). ^13^C NMR (176 MHz, CDCl_3_): δ 22.8 (CH_3_), 120.0 (C-3a), 121.1 (C-7), 121.3 (NPh C-2,6), 121.9 (C-3), 127.1 (C7Ph C-4), 127.2 (C6Ph C-4), 127.6 (C7Ph C-1), 127.8 (C7Ph), 127.9 (C6Ph C-3,5), 128.6 (NPh C-4), 129.6 NPh C-3,5), 130.6 (C7Ph), 131.2 (C6Ph C-2,6), 135.7 (C6Ph C-1), 140.0 (NPh C-1), 148.8 (C-6), 151.5 (C-7a), 154.5 (C-4). ^15^N NMR (71 MHz, CDCl_3_): δ −145.9 (N-2), −95.0 (N-1), −88.2 (N-5). IR (KBr, ν, cm^−1^): 3137, 3061 (CH_arom_), 2916 (CH_aliph_), 1588, 1545, 1505, 1476, 1371 (C=C, C=N, C−N), 762, 700, 661 (CH=CH monosubstituted benzenes). MS (ES^+^): *m/z* (%): 362 ([M + H]^+^, 100). HRMS (ESI) C_25_H_20_N_3_ ([M + H]^+^): requires 362.1652 and found 362.1650.

##### 7-(2-Methoxyphenyl)-4-methyl-2,6-diphenyl-2*H*-pyrazolo[4,3-*c*]pyridine **25**

7-(2-Methoxyphenyl)-4-methyl-2,6-diphenyl-2*H*-pyrazolo[4,3-*c*]pyridine **25** was prepared in accordance with general procedure (D) from 7-iodo-4-methyl 2,6-diphenyl-2*H*-pyrazolo[4,3-*c*]pyridine **14** (50 mg, 0.12 mmol), (2-methoxyphenyl)boronic acid (22 mg, 0.145 mmol), Cs_2_CO_3_ (79 mg, 0.24 mmol), Pd(OAc)_2_ (1.9 mg, 0.008 mmol), EtOH (0.9 mL), and water (0.3 mL). The reaction was finished after 1 h. The desired compound was obtained after purification by column chromatography (EtOAc/Hex, 1:3 *v*/*v*). Yield: 38 mg (80%), light yellow crystalline solid, mp = 159–160 °C, R*_f_* = 0.13 (EtOAc/Hex, 1:3 *v*/*v*). ^1^H NMR (500 MHz, CDCl_3_): δ 2.95 (3H, s, CH_3_), 3.42 (3H, s, OCH_3_), 6.81–6.85 (1H, m, C7Ph 3-H), 6.97–7.01 (1H, m, C6Ph 4-H), 7.17–7.23 (3H, m, NPh 3,5-H and C6Ph 4-H), 7.28–7.32 (1H, m, C7Ph 4-H), 7.39–7.45 (4H, m, NPh 4-H, C6Ph 2,6-H and C7Ph 5-H), 7.47–7.51 (2H, m, C6Ph 3,5-H), 7.85–7.89 (2H, m, NPh, 2,6-H), 8.62 (1H, s, 3-H). ^13^C NMR (126 MHz, CDCl_3_): δ 22.9 (CH_3_), 55.4 (OCH_3_), 111.5 (C7Ph C-3), 117.8 (C-7), 120.1 (C-3a), 120.6 (C7Ph C-5), 121.6 (NPh C-2,6), 122.2 (C-3), 125.0 (C7Ph C-1), 127.2 (C6Ph C-4), 127.6 (NPh C-3,5), 128.7 (NPh C-4), 129.2 (C7Ph C-4), 129.7 (C6Ph C-2,3,5,6), 132.6 (C7Ph C-5), 140.3 (NPh C-1), 140.8 (C6Ph C-1), 149.8 (C-6), 151.8 (C-7a), 154.8 (C-4), 157.1 (C7Ph C-2). IR (ν, cm^−1^): 3143, 3069, 3019 (CH_arom_), 2955, 2922, 2853 (CH_aliph_), 1599, 1586, 1578, 1552, 1504, 1495, 1479, 1434, 1372, 1240, 1217 (C=C, C=N, C–N), 1046, 1022 (C-O-C), 750, 698, 685 (CH=CH of mono- and disubstituted benzenes). MS (ES^+^): *m/z* (%): 392 ([M + H]^+^, 100). HRMS (ESI) for C_26_H_22_N_3_O ([M + H]^+^): requires 392.1758 and found 392.1757.

##### 7-(3-Methoxyphenyl)-4-methyl-2,6-diphenyl-2*H*-pyrazolo[4,3-*c*]pyridine **26**

7-(3-Methoxyphenyl)-4-methyl-2,6-diphenyl-2*H*-pyrazolo[4,3-*c*]pyridine **26** was prepared in accordance with general procedure (D) from 7-iodo-4-methyl 2,6-diphenyl-2*H*-pyrazolo[4,3-*c*]pyridine **14** (50 mg, 0.12 mmol), (3-methoxyphenyl)boronic acid (22 mg, 0.145 mmol), Cs_2_CO_3_ (79 mg, 0.24 mmol), Pd(OAc)_2_ (1.9 mg, 0.008 mmol), EtOH (0.9 mL), and water (0.3 mL). The reaction was finished after 30 min. The desired compound was obtained after purification by column chromatography (EtOAc/Hex, 1:3 *v*/*v*). Yield: 38 mg (80%), light yellow crystalline solid, mp = 190–191 °C, R*_f_* = 0.18 (EtOAc/Hex, 1:3 *v*/*v*). ^1^H NMR (500 MHz, CDCl_3_): δ 2.97 (3H, s, CH_3_), 3.65 (3H, s, OCH_3_), 6.79–6.85 (1H, m, C7Ph 4-H), 6.96–7.03 (1H, m, C7Ph 2-H), 7.06–7.12 (1H, m, C7Ph 6-H), 7.17–7.32 (4H, m, C7Ph 5-H, C6Ph 3,4,5-H), 7.41–7.47 (1H, m, NPh 4-H), 7.50–7.55 (4H, m, NPh 3,5-H, C6Ph, 2,6-H), 7.86–7.95 (2H, m, NPh 2,6-H), 8.67 (1H, s, 3-H). ^13^C NMR (126 MHz, CDCl_3_): δ 23.1 (CH_3_), 55.2 (OCH_3_), 113.4 (C7Ph C-4), 116.6 (C7Ph C-2), 120.1 (C-3a), 120.8 (C-7), 121.3 (NPh C-2,6), 121.9 (C-3), 123.9 (C7Ph C-6), 127.2 (C6Ph C-4), 127.9 (C6Ph C-3,5), 128.6 (NPh, C-4), 129.0 (C7Ph C-5), 129.7 (NPh C-3,5), 130.6 (C6Ph C-2,6), 137.2 (C7Ph C-1), 140.1 (NPh C-1), 140.8 (C6Ph C-1), 149.3 (C-6), 151.4 (C-7a), 154.6 (C-4), 159.2 (C7Ph C-3). IR (ν, cm^−1^): 3067, 3002 (CH_arom_), 2954, 2923, 2852 (CH_aliph_), 1599, 1589, 1575, 1547, 1488, 1443, 1284, 1211 (C=C, C=N, C–N), 1160, 1049 (C-O-C), 753, 700, 686 (CH=CH of mono- and disubstituted benzenes). MS (ES^+^): *m/z* (%): 392 ([M + H]^+^, 100). HRMS (ESI) for C_26_H_22_N_3_O ([M + H]^+^): requires 392.1757 and found 392.1757.

##### 7-(4-Methoxyphenyl)-4-methyl-2,6-diphenyl-2*H*-pyrazolo[4,3-*c*]pyridine **27**

7-(4-Methoxyphenyl)-4-methyl-2,6-diphenyl-2*H*-pyrazolo[4,3-*c*]pyridine **27** was prepared in accordance with general procedure (D) from 7-iodo-4-methyl 2,6-diphenyl-2*H*-pyrazolo[4,3-*c*]pyridine **14** (50 mg, 0.12 mmol), (4-methoxyphenyl)boronic acid (22 mg, 0.145 mmol), Cs_2_CO_3_ (79 mg, 0.24 mmol), Pd(OAc)_2_ (1.9 mg, 0.008 mmol), EtOH (0.9 mL), and water (0.3 mL). The reaction was finished after 30 min. The desired compound was obtained after purification by column chromatography (EtOAc/Hex, 1:3 *v*/*v*). Yield: 118 mg (89%), mp = 157–161 °C, R*_f_* = 0.15 (EtOAc/Hex, 1:3 *v*/*v*). ^1^H NMR (700 MHz, CDCl_3_): δ 2.92 (3H, s, CH_3_), 3.84 (3H, s, OCH_3_), 6.88–6.89 (2H, m, C7Ph 3,5-H), 7.23–7.25 (1H, m, C6Ph 4-H), 7.27–7.30 (2H, m, C6Ph 3,5-H), 7.42–7.45 (3H, m, NPh 4-H and C7Ph 2,6-H), 7.49–7.50 (2H, m, C6Ph 2,6-H), 7.52–7.54 (2H, m, NPh 3,5-H), 7.92–7.93 (2H, m, NPh 2,6-H), 8.61 (1H, s, 3-H). ^13^C NMR (176 MHz, CDCl_3_): δ 23.2 (CH_3_), 55.3 (OCH_3_), 113.6 (C7Ph C-3,5), 120.2 (C-3a), 120.5 (C-7), 121.4 (NPh C-2,6), 121.7 (C-3), 127.0 (C6Ph C-4), 127.9 (C6Ph C-3,5), 128.3 (NPh C-4), 128.5 (C7Ph C-1), 129.7 (NPh C-3,5), 130.7 (C6Ph C-2,6), 132.5 (C7Ph C-2,6), 140.3 (NPh C-1), 141.3 (C6Ph C-1), 149.2 (C-6), 151.8 (C-7a), 154.0 (C-4), 158.7 (C7Ph C-4). ^15^N NMR (71 MHz, CDCl_3_): δ −146.8 (N-2), −95.7 (N-1), −81.6 (N-5). IR (KBr, ν, cm^−1^): 3056, 3012 (CH_arom_), 2951, 2834, 2903, 2831 (CH_aliph_), 1607, 1598, 1588, 1507, 1247 (C=C, C=N, C–N), 1075, 1038 (C–O), 755, 728, 701, 685 (CH=CH of mono- and disubstituted benzenes). MS (ES^+^): *m/z* (%): 392 ([M + H]^+^, 100). HRMS (ESI) C_26_H_22_N_3_O ([M + H]^+^): requires 392.1757 and found 392.1759.

##### 7-(3,4-Dimethoxyphenyl)-4-methyl-2,6-diphenyl-2*H*-pyrazolo[4,3-*c*]pyridine **28**

7-(3,4-Dimethoxyphenyl)-4-methyl-2,6-diphenyl-2*H*-pyrazolo[4,3-*c*]pyridine **28** was prepared in accordance with general procedure (D) from 7-iodo-4-methyl 2,6-diphenyl-2*H*-pyrazolo[4,3-*c*]pyridine **14** (50 mg, 0.12 mmol), (3,4-dimethoxyphenyl)boronic acid (26 mg, 0.145 mmol), Cs_2_CO_3_ (79 mg, 0.24 mmol), Pd(OAc)_2_ (1.9 mg, 0.008 mmol), EtOH (0.9 mL), and water (0.3 mL). The reaction was finished after 1 h. The desired compound was obtained after purification by column chromatography (EtOAc/Hex, 1:3 *v*/*v*). Yield: 41 mg (67%), light yellow crystalline solid, mp = 180–181 °C, R*_f_* = 0.12 (EtOAc/Hex, 1:3 *v*/*v*). ^1^H NMR (700 MHz, CDCl_3_): δ 2.91 (3H, s, CH_3_), 3.60 (3H, s, 3-OCH_3_), 3.89 (3H, s, 4-OCH_3_), 6.85–6.90 (2H, m, C7Ph 2,5-H), 7.20–7.24 (2H, m, C7Ph 6-H, C6Ph 4-H), 7.24–7.28 (2H, m, C6Ph 3,5-H), 7.41–7.44 (1H, m, NPh 4-H), 7.44–7.47 (2H, m, C6Ph 2,6-H), 7.50–7.54 (2H, m, NPh 3,5-H), 7.90–7.95 (2H, m, NPh 2,6-H), 8.62 (1H, s, 3-H). ^13^C NMR (176 MHz, CDCl_3_): δ 23.2 (CH_3_), 55.7 (3-OCH_3_), 55.9 (4-OCH_3_), 110.8 (C7Ph C-5), 114.9 (C7Ph C-2), 120.3 (C-3a), 120.6 (C-7), 121.4 (NPh C-2,6), 121.8 (C-3), 123.9 (C7Ph C-6), 127.1 (C6Ph C-4), 128.1 (C6Ph C-3,5), 128.3 (C7Ph C-1), 128.6 (NPh C-4), 129.7 (NPh C-3,5), 130.6 (C6Ph C-2,6), 140.3 (NPh C-1), 141.3 (CPh C-1), 148.2 (C7Ph C-4), 148.3 (C7Ph C-3), 149.3 (C-6), 151.6 (C-7a), 154.2 (C-4). ^15^N NMR (71 MHz, CDCl_3_): δ −147.2 (N-2), −96.4 (N-1), −82.9 (N-5). IR (ν, cm^−1^): 3121, 3049, 2999 (CH_arom_), 2987, 2949, 2937, 2832 (CH_aliph_), 1589, 1519, 1508, 1481, 1465, 1256, 1231 (C=C, C=N, C–N), 1164, 1138, 1023 (C-O-C), 759, 728, 701, 686, 667 (CH=CH of mono- and trisubstituted benzenes). MS (ES^+^): *m/z* (%): 422 ([M + H]^+^, 98). HRMS (ESI) for C_27_H_24_N_3_O_2_ ([M + H]^+^): requires 422.1863 and found 422.1863.

##### 4-(4-Methyl-2,6-diphenyl-2*H*-pyrazolo[4,3-*c*]pyridin-7-yl)phenol **29**

4-(4-Methyl-2,6-diphenyl-2*H*-pyrazolo[4,3-*c*]pyridin-7-yl)phenol **29** was prepared in accordance with general procedure (D) from -iodo-4-methyl 2,6-diphenyl-2*H*-pyrazolo[4,3-*c*]pyridine **14** (80 mg, 0.2 mmol), (4-hydroxyphenyl)boronic acid (32 mg, 0.23 mmol), Cs_2_CO_3_ (127 mg, 0.39 mmol), Pd(OAc)_2_ (3 mg, 0.014 mmol), EtOH (1.2 mL), and water (0.4 mL). The reaction was finished after 30 min. The desired compound was obtained after purification by column chromatography (EtOAc/Hex, 1:4 to 1:2 *v*/*v*). Yield: 37 mg (50%), yellowish crystalline solid, mp = 269–270 °C, R*_f_* = 0.20 (EtOAc/Hex, 1:2 *v*/*v*). ^1^H NMR (700 MHz, DMSO-*d*_6_): δ 2.80 (3H, s, CH_3_), 6.68–6.73 (2H, m, C7Ph 3,5-H), 7.16–7.19 (2H, m, C7Ph 2,6-H), 7.20–7.22 (1H, m, C6Ph 4-H), 7.22–7.26 (2H, m, C6Ph 3,5-H), 7.34–7.38 (2H, m, C6Ph 2,6-H), 7.46–7.49 (1H, m, NPh 4-H), 7.57–7.63 (2H, m, NPh 3,5-H), 8.04–8.08 (2H, m, NPh 2,6-H), 9.44 (1H, s, OH), 9.51 (1H, s, 3-H). ^13^C NMR (176 MHz, DMSO-*d*_6_): δ 22.6 (CH_3_), 114.9 (C7Ph C-3,5), 119.7 (C-3a), 120.0 (C-7), 120.6 (NPh C-2,6), 123.9 (C-3), 126.3 (C7Ph C-1), 126.7 (C6Ph C-4), 127.4 (C6Ph C-3,5), 128.4 (NPh C-4), 129.7 (NPh C-3,5), 130.3 (C6Ph C-2,6), 132.0 (C7Ph C-2,6), 139.6 (NPh C-1), 141.1 (C6Ph C-1), 148.0 (C-6), 150.9 (C-7a), 153.9 (C-4), 156.4 (C7Ph C-4). ^15^N NMR (71 MHz, DMSO-*d*_6_): δ −147.2 (N-2), −80.5 (N-5), N-1 not found. IR (KBr, ν, cm^−1^): 3455 (OH), 3149 (CH_arom_), 2920 (CH_aliph_), 1609, 1591, 1509, 1397, 1264 (C=C, C=N, C–N), 828, 757, 698 (CH=CH of mono- and disubstituted benzenes). MS (ES^+^): *m/z* (%): 378 ([M + H]^+^, 97). HRMS (ESI) for C_25_H_20_N_3_O ([M + H]^+^): requires 378.1601 and found 378.1602.

##### 4-Ethyl-2,6,7-triphenyl-2*H*-pyrazolo[4,3-*c*]pyridine **30**

4-Ethyl-2,6,7-triphenyl-2*H*-pyrazolo[4,3-*c*]pyridine **30** was prepared in accordance with general procedure (D) from 7-iodo-4-methyl 2,6-diphenyl-2*H*-pyrazolo[4,3-*c*]pyridine **15** (80 mg, 0.19 mmol), phenylboronic acid (31 mg, 0.23 mmol), Cs_2_CO_3_ (123 mg, 0.38 mmol), Pd(OAc)_2_ (3 mg, 0.013 mmol), EtOH (1.2 mL), and water (0.4 mL). The reaction was finished after 30 min. The desired compound was obtained after purification by column chromatography (EtOAc/Hex, 1:5 *v*/*v*). Yield: 58 mg (81%), yellowish crystalline solid, mp = 179–180 °C, R*_f_* = 0.39 (EtOAc/Hex, 1:3 *v*/*v*). ^1^H NMR (700 MHz, CDCl_3_): δ 1.54 (3H, t, *J* = 7.6 Hz, CH_3_), 3.23 (2H, q, *J* = 7.6 Hz, CH_2_), 7.19–7.23 (1H, m, C6Ph 4-H), 7.23–7.26 (2H, m, C6Ph 3,5-H), 7.26–7.29 (1H, m, C7Ph 4-H), 7.29–7.34 (2H, m, C7Ph 3,5-H), 7.40–7.44 (1H, m, NPh 4-H), 7.46–7.50 (4H, m, CPh 2,6-H), 7.50–7.53 (2H, m, NPh 3,5-H), 7.87–7.95 (2H, m, NPh 2,6-H), 8.62 (1H, s, 3-H). ^13^C NMR (176 MHz, CDCl_3_): δ 13.3 (CH_3_), 30.4 (CH_2_), 119.0 (C-3a), 120.7 (C-7), 121.2 (C-4), 121.3 (NPh C-2,6), 127.0 (C6Ph and C7Ph C-4), 127.7 (C6Ph C-3,5), 127.9 (C7Ph C-3,5), 128.4 (NPh C-4), 129.5 (NPh C-3,5), 130.7 (C6Ph C-2,6), 131.2 (C7Ph C-2,6), 136.2 (C7Ph C-1), 140.2 (NPh C-1), 141.0 (C6Ph C-1), 149.1 (C-6), 151.9 (C-7a), 159.3 (C-4). ^15^N NMR (71 MHz, CDCl_3_): δ −147.3 (N-2), N-1 not found, −83.4 (N-5). IR (KBr, ν, cm^−1^): 3059 (CH_arom_), 2970, 2930 (CH_aliph_), 1587, 1547, 1476, 1371, 1213 (C=C, C=N, C−N), 753, 697, 686 (CH=CH monosubstituted benzenes). MS (ES^+^): *m/z* (%): 375 ([M + H]^+^, 99). HRMS (ESI) for C_26_H_22_N_3_ ([M + H]^+^): requires 376.1808 and found 376.1808.

##### 4-Ethyl-7-(4-methoxyphenyl)-2,6-diphenyl-2*H*-pyrazolo[4,3-*c*]pyridine **31**

4-Ethyl-7-(4-methoxyphenyl)-2,6-diphenyl-2*H*-pyrazolo[4,3-*c*]pyridine **31** was prepared in accordance with general procedure (D) from 7-iodo-4-methyl 2,6-diphenyl-2*H*-pyrazolo[4,3-*c*]pyridine **15** (80 mg, 0.19 mmol), (4-methoxyphenyl)boronic acid (34 mg, 0.23 mmol), Cs_2_CO_3_ (123 mg, 0.38 mmol), Pd(OAc)_2_ (3 mg, 0.013 mmol), EtOH (1.2 mL), and water (0.4 mL). The reaction was finished after 30 min. The desired compound was obtained after purification by column chromatography (EtOAc/Hex, 1:5 *v*/*v*). Yield: 63 mg (82%), yellowish crystalline solid, mp = 161–162 °C, R*_f_* = 0.34 (EtOAc/Hex, 1:3 *v*/*v*). ^1^H NMR (700 MHz, CDCl_3_): δ 1.53 (3H, t, *J* = 7.6 Hz, CH_3_), 3.21 (2H, q, *J* = 7.6 Hz, CH_2_), 3.82 (3H, s, OCH_3_), 6.83–6.90 (2H, m, C7Ph 3,5-H), 7.19–7.23 (1H, m, C6Ph 4-H), 7.24-7.27 (2H, m, C6Ph 3,5-H), 7.40–7.44 (3H, m, C7Ph 2,6-H and NPh 4-H), 7.47–7.50 (2H, m, C6Ph 2,6-H), 7.50–7.53 (2H, m, NPh 3,5-H), 7.86–7.97 (2H, m, NPh 2,6-H), 8.61 (1H, s, 3-H). ^13^C NMR (176 MHz, CDCl_3_): δ 13.3 (CH_3_), 30.4 (CH_2_), 55.1 (OCH_3_), 113.5 (C7Ph C-3,5), 119.0 (C-3a), 120.3 (C-7), 121.2 (C-4), 121.3 (NPh C-2,6), 126.9 (C6Ph C-4), 127.7 (C6Ph C-3,5), 128.3 (C7Ph C-1), 128.4 (NPh C-4), 129.5 (NPh C-3,5), 130.6 (C6Ph C-2,6), 132.3 (C7Ph C-2,6), 140.2 (NPh C-1), 141.2 (C6Ph C-1), 148.9 (C-6), 152.1 (C-7a), 158.6 (C7Ph C-4), 158.9 (C-4). ^15^N NMR (71 MHz, CDCl_3_): δ −147.6 (N-2), N-1 not found, −83.2 (N-5). IR (KBr, ν, cm^−1^): 3058, 3016 (CH_arom_), 2986, 2934, 2889 (CH_aliph_), 1607, 1507, 1462, 1376, 1290, 1248, 1176 (C=C, C=N, C–N), 1045 (C–O), 830, 753, 699, 686 (CH=CH of mono- and disubstituted benzenes). MS (ES^+^): *m/z* (%): 405 ([M + H]^+^, 99). HRMS (ESI) for C_27_H_24_N_3_O ([M + H]^+^): requires 406.1914 and found 406.1914.

##### 7-(2,4-Dimethoxyphenyl)-4-ethyl-2,6-diphenyl-2*H*-pyrazolo[4,3-*c*]pyridine **32**

7-(2,4-Dimethoxyphenyl)-4-ethyl-2,6-diphenyl-2*H*-pyrazolo[4,3-*c*]pyridine **32** was prepared in accordance with general procedure (D) from 7-iodo-4-methyl 2,6-diphenyl-2*H*-pyrazolo[4,3-*c*]pyridine **15** (80 mg, 0.19 mmol), (2,4-dimethoxyphenyl)boronic acid (41 mg, 0.23 mmol), Cs_2_CO_3_ (123 mg, 0.38 mmol), Pd(OAc)_2_ (3 mg, 0.013 mmol), EtOH (1.2 mL), and water (0.4 mL). The reaction was finished after 30 min. The desired compound was obtained after purification by column chromatography (EtOAc/Hex, 1:5 *v*/*v*). Yield: 40 mg (48%), yellow crystalline solid, mp = 202–203 °C, R*_f_* = 0.24 (EtOAc/Hex, 1:3 *v*/*v*). ^1^H NMR (700 MHz, CDCl_3_): δ 1.53 (3H, t, *J* = 7.6 Hz, CH_3_), 3.21 (2H, q, *J* = 7.6 Hz, CH_2_), 3.37 (3H, s, 2-OCH_3_), 3.83 (3H, s, 4-OCH_3_), 6.38–6.44 (1H, m, C7Ph 3-H), 6.53–6.59 (1H, m, C7Ph 5-H), 7.15–7.19 (1H, m, C6Ph 4-H), 7.19–7.25 (2H, m, C6Ph 3,5-H), 7.35–7.41 (2H, m, NPh 4-H and C7Ph 6-H), 7.45–7.51 (4H, m, C6Ph 2,6-H and NPh 3,5-H), 7.84–7.90 (2H, m, NPh 2,6-H), 8.58 (1H, s, 3-H). ^13^C NMR (176 MHz, CDCl_3_): δ 13.3 (CH_3_), 30.4 (CH_2_), 55.1 (2-OCH_3_), 55.3 (4-OCH_3_), 99.1 (C7Ph C-3), 104.6 (C7Ph C-5), 116.8 (C-7), 117.9 (C7Ph C-1), 119.0 (C-3a), 121.2 (C-3), 121.4 (NPh C-2,6), 126.7 (C6Ph C-4), 127.4 (C6Ph C-3,5), 128.3 (NPh C-4), 129.5 (NPh C-3,5 and C6Ph C-2,6), 132.8 (C7Ph C-6), 140.3 (NPh C-1), 141.9 (C6Ph C-1), 149.9 (C-6), 152.4 (C-7a), 157.9 (C7Ph C-2), 159.0 (C-4), 160.5 (C7Ph C-4). ^15^N NMR (71 MHz, CDCl_3_): δ −147.9 (N-2), −96.4 (N-1), −84.6 (N-5). IR (KBr, ν, cm^−1^): 3134, 3058 (CH_arom_), 2966, 2930, 2836 (CH_aliph_), 1606, 1588, 1547, 1505, 1462, 1305, 1204 (C=C, C=N, C–N), 1027 (C–O), 829, 764, 705, 691 (CH=CH of mono- and trisubstituted benzenes). MS (ES^+^): *m/z* (%): 435 ([M + H]^+^, 100). HRMS (ESI) for C_28_H_26_N_3_O_2_ ([M + H]^+^): requires 436.2020 and found 436.2020.

##### 4-Ethyl-2,6-diphenyl-7-(*p*-tolyl)-2*H*-pyrazolo[4,3-*c*]pyridine **33**

4-Ethyl-2,6-diphenyl-7-(*p*-tolyl)-2*H*-pyrazolo[4,3-*c*]pyridine **33** was prepared in accordance with general procedure (D) from 7-iodo-4-methyl 2,6-diphenyl-2*H*-pyrazolo[4,3-*c*]pyridine **15** (80 mg, 0.19 mmol), (4-methylphenyl)boronic acid (31 mg, 0.23 mmol), Cs_2_CO_3_ (123 mg, 0.38 mmol), Pd(OAc)_2_ (3 mg, 0.013 mmol), EtOH (1.2 mL), and water (0.4 mL). The reaction was finished after 30 min. The desired compound was obtained after purification by column chromatography (EtOAc/Hex, 1:8 *v*/*v*). Yield: 60 mg (81%), yellowish crystalline solid, mp = 179–180 °C, R*_f_* = 0.41 (EtOAc/Hex, 1:3 *v*/*v*). ^1^H NMR (700 MHz, CDCl_3_): δ 1.52 (3H, t, *J* = 7.6 Hz, CH_2_C*H*_3_), 2.34 (3H, s, Ph-C*H*_3_), 3.19 (2H, q, *J* = 7.6 Hz, C*H*_2_CH_3_), 7.09–7.14 (2H, m, C7Ph 3,5-H), 7.19–7.22 (1H, m, C6Ph 4-H), 7.23–7.25 (2H, m, C6Ph 3,5-H), 7.34–7.38 (2H, m, C6Ph 2,6-H), 7.38–7.41 (1H, m, NPh 4-H), 7.46–7.51 (4H, m, NPh 3,5-H and C7Ph 2,6-H), 7.86–7.91 (2H, m, NPh 2,6-H), 8.59 (1H, s, 3-H). ^13^C NMR (176 MHz, CDCl_3_): δ 13.3 (CH_2_*C*H_3_), 21.3 (Ph-CH_3_), 30.4 (*C*H_2_CH_3_), 119.0 (C-3a), 120.7 (C-7), 121.2 (C-4), 121.3 (NPh C-2,6), 126.9 (C6Ph C-4), 127.7 (C6Ph C-3,5), 128.3 (NPh C-4), 128.7 (C7Ph C-3), 129.5 (NPh C-3,5), 130.6 (C7Ph C-2,6), 131.0 (C6Ph C-2,6), 133.0 (C7Ph C-1), 136.5 (C7Ph C-4), 140.2 (NPh C-1), 141.2 (C6Ph C-1), 148.9 (C-6), 152.0 (C-7a), 159.0 (C-4). ^15^N NMR (71 MHz, CDCl_3_): δ −147.5 (N-2), −96.1 (N-1), −83.3 (N-5). IR (KBr, ν, cm^−1^): 3028 (CH_arom_), 2990, 2939 (CH_aliph_), 1587, 1505, 1463, 1377, 1211, 1045 (C=C, C=N, C–N), 822, 753, 699, 686 (CH=CH of mono- and disubstituted benzenes). MS (ES^+^): *m/z* (%): 389 ([M + H]^+^, 98). HRMS (ESI) for C_27_H_24_N_3_ ([M + H]^+^): requires 390.1965 and found 390.1965.

##### 4-Ethyl-2,6-diphenyl-7-[4-(trifluoromethyl)phenyl]-2*H*-pyrazolo[4,3-*c*]pyridine **34**

4-Ethyl-2,6-diphenyl-7-[4-(trifluoromethyl)phenyl]-2*H*-pyrazolo[4,3-*c*]pyridine **34** was prepared in accordance with general procedure (D) from 7-iodo-4-methyl 2,6-diphenyl-2*H*-pyrazolo[4,3-*c*]pyridine **15** (80 mg, 0.19 mmol), 4-(trifluoromethyl)phenylboronic acid (43 mg, 0.23 mmol), Cs_2_CO_3_ (123 mg, 0.38 mmol), Pd(OAc)_2_ (3 mg, 0.013 mmol), EtOH (1.2 mL), and water (0.4 mL). The reaction was finished after 30 min. The desired compound was obtained after purification by column chromatography (EtOAc/Hex, 1:8 *v*/*v*). Yield: 77 mg (92%), yellowish crystalline solid, mp = 203–204 °C, R*_f_* = 0.39 (EtOAc/Hex, 1:3 *v*/*v*). ^1^H NMR (700 MHz, CDCl_3_): δ 1.54 (3H, t, *J* = 7.6 Hz, CH_3_), 3.19 (2H, q, *J* = 7.6 Hz, CH_2_), 7.22–7.28 (3H, m, C6Ph 3,4,5-H), 7.41–7.46 (3H, m, C6Ph 2,6-H, NPh 4-H), 7.50–7.54 (2H, m, NPh 3,5-H), 7.55–7.58 (2H, m, C7Ph 3,5-H), 7.59–7.63 (2H, m, C7Ph 2,6-H), 7.88–7.92 (2H, m, NPh 2,6-H), 8.64 (1H, s, 3-H). ^13^C NMR (176 MHz, CDCl_3_): δ 13.3 (CH_3_), 30.4 (CH_2_), 119.0 (C-3a), 119.2 (C-7), 121.3 (NPh C-2,6), 121.5 (C-4), 124.3 (CF_3_, *J* = 201.6 Hz), 124.8 (C7Ph C-3,5, *J* = 2.52 Hz), 127.4 (C6Ph C-4), 127.9 (C6Ph C-3,5), 128.6 (NPh C-4), 128.8 (C7Ph C-4, *J* = 25.2 Hz), 129.6 (NPh C-3,5), 130.7 (C6Ph C-2,6), 131.5 (C7Ph C-2,6), 140.0 (NPh C-1), 140.1 (C7Ph C-1), 140.4 (C6Ph C-1), 149.6 (C-6), 151.5 (C-7a), 160.1 (C-4). ^15^N NMR (71 MHz, CDCl_3_): δ −146.9 (N-2), −97.4 (N-1), −83.5 (N-5). ^19^F NMR (376 MHz, CDCl_3_): δ −65.59 (3F, s, CF_3_). IR (KBr, ν, cm^−1^): 3053 (CH_arom_), 2971 (CH_aliph_), 1585, 1484, 1327, 1130 (C=C, C=N, C–N, C-F), 759, 696 (CH=CH of mono- and disubstituted benzenes). MS (ES^+^): *m/z* (%): 443 ([M + H]^+^, 99). HRMS (ESI) for C_27_H_21_F_3_N_3_ ([M + H]^+^): requires 444.1682 and found 444.1682.

##### 4-Ethyl-2,6-diphenyl-7-[4-(trifluoromethoxy)phenyl]-2*H*-pyrazolo[4,3-*c*]pyridine **35**

4-Ethyl-2,6-diphenyl-7-[4-(trifluoromethoxy)phenyl]-2*H*-pyrazolo[4,3-*c*]pyridine **35** was prepared in accordance with general procedure (D) from 7-iodo-4-methyl 2,6-diphenyl-2*H*-pyrazolo[4,3-*c*]pyridine **15** (80 mg, 0.19 mmol), 4-(trifluoromethoxy)phenylboronic acid (47 mg, 0.23 mmol), Cs_2_CO_3_ (123 mg, 0.38 mmol), Pd(OAc)_2_ (3 mg, 0.013 mmol), EtOH (1.2 mL), and water (0.4 mL). The reaction was finished after 30 min. The desired compound was obtained after purification by column chromatography (EtOAc/Hex, 1:8 *v*/*v*). Yield: 74 mg (85%), white crystalline solid, mp = 153–154 °C, R*_f_* = 0.41 (EtOAc/Hex, 1:3 *v*/*v*). ^1^H NMR (700 MHz, CDCl_3_): δ 1.54 (3H, t, *J* = 7.6 Hz, CH_3_), 3.22 (2H, q, *J* = 7.6 Hz, CH_2_), 7.12–7.19 (2H, m, C7Ph 3,5-H), 7.22–7.28 (3H, m, C6Ph 3,4,5-H), 7.40–7.48 (3H, m, C6Ph 2,6-H, NPh 4-H), 7.49–7.56 (4H, m, NPh 3,5-H and C7Ph C-2,6), 7.87–7.94 (2H, m, NPh 2,6-H), 8.63 (1H, s, 3-H). ^13^C NMR (176 MHz, CDCl_3_): δ 13.3 (CH_3_), 30.4 (CH_2_), 119.0 (C-3a), 119.2 (C-7), 120.3 (C7Ph C-3,5), 120.5 (CF_3_, *J*=176.4 Hz), 121.3 (NPh C-2,6), 121.5 (C-4), 127.2 (C6Ph C-4), 127.8 (C6Ph C-3,5), 128.6 (NPh C-4), 129.6 (NPh C-3,5), 130.6 (C6Ph C-2,6), 132.6 (C7Ph C-2,6), 134.8 (C7Ph C-4), 140.1 (NPh C-1), 140.6 (C6Ph C-1), 148.2 (C7Ph C-1), 149.4 (C-6), 151.6 (C-7a), 159.8 (C-4). ^15^N NMR (71 MHz, CDCl_3_): δ −147.1 (N-2), −97.4 (N-1), −83.5 (N-5). ^19^F NMR (376 MHz, CDCl_3_): δ −60.82 (3F, s, CF_3_). IR (KBr, ν, cm^−1^): 3049 (CH_arom_), 2969, 2934 (CH_aliph_), 1586, 1507, 1367, 1259, 1225, 1168 (C=C, C=N, C–N, C-F), 757, 699, 687 (CH=CH of mono- and disubstituted benzenes). MS (ES^+^): *m/z* (%): 459 ([M + H]^+^, 99). HRMS (ESI) for C_27_H_21_F_3_N_3_O ([M + H]^+^): requires 460.163 and found 460.1631.

##### 7-(4-Chlorophenyl)-4-ethyl-2,6-diphenyl-2*H*-pyrazolo[4,3-*c*]pyridine **36**

7-(4-Chlorophenyl)-4-ethyl-2,6-diphenyl-2*H*-pyrazolo[4,3-*c*]pyridine **36** was prepared in accordance with general procedure (D) from 7-iodo-4-methyl 2,6-diphenyl-2*H*-pyrazolo[4,3-*c*]pyridine **15** (80 mg, 0.19 mmol), 4-chlorophenylboronic acid (36 mg, 0.23 mmol), Cs_2_CO_3_ (123 mg, 0.38 mmol), Pd(OAc)_2_ (3 mg, 0.013 mmol), EtOH (1.2 mL), and water (0.4 mL). The reaction was finished after 30 min. The desired compound was obtained after purification by column chromatography (EtOAc/Hex, 1:8 *v*/*v*). Yield: 65 mg (83%), yellow crystalline solid, mp = 226–227 °C, R*_f_* = 0.49 (EtOAc/Hex, 1:3 *v*/*v*). ^1^H NMR (700 MHz, CDCl_3_): δ 1.52 (3H, t, *J* = 7.6 Hz, CH_3_), 3.21 (2H, q, *J* = 7.6 Hz, CH_2_), 7.22–7.30 (5H, m, C6Ph 3,4,5-H; C7Ph 3,5-H), 7.40–7.44 (3H, m, C7Ph 2,6-H; NPh 4-H), 7.44–7.47 (2H, m, C6Ph 2,6-H), 7.50–7.53 (2H, m, NPh 3,5-H), 7.86–7.93 (2H, m, NPh 2,6-H), 8.61 (1H, s, 3-H). ^13^C NMR (176 MHz, CDCl_3_): δ 13.3 (CH_3_), 30.4 (CH_2_), 119.0 (C-3a), 119.4 (C-7), 121.3 (NPh C-2,6), 121.4 (C-3), 127.2 (C6Ph C-4), 127.9 (C6Ph C-3,5), 128.2 (C7Ph C-3,5), 128.5 (NPh C-4), 129.6 (NPh C-3,5), 130.6 (C6Ph C-2,6), 132.5 (C7Ph C-2,6), 132.8 (C7Ph C-4), 134.6 (C7Ph C-1), 140.1 (NPh C-1), 140.6 (C6Ph C-1), 149.3 (C-6), 151.6 (C-7a), 159.7 (C-4). ^15^N NMR (71 MHz, CDCl_3_): δ −147.3 (N-2), −97.0 (N-1), −83.3 (N-5). IR (KBr, ν, cm^−1^): 3057 (CH_arom_), 2991, 2939 (CH_aliph_), 1691, 1587, 1501, 1463, 1213, 1092 (C=C, C=N, C–N), 825, 756, 698, 688 (CH=CH of mono- and disubstituted benzenes). MS (ES^+^): *m/z* (%): 409 ([M + H]^+^, 96). HRMS (ESI) for C_26_H_21_ClN_3_ ([M + H]^+^): requires 410.1419 and found 410.1419.

##### 4-(4-Ethyl-2,6-diphenyl-2*H*-pyrazolo[4,3-*c*]pyridin-7-yl)phenol **37**

4-(4-Ethyl-2,6-diphenyl-2*H*-pyrazolo[4,3-*c*]pyridin-7-yl)phenol **37** was prepared in accordance with general procedure (D) from 7-iodo-4-methyl 2,6-diphenyl-2*H*-pyrazolo[4,3-*c*]pyridine **15** (80 mg, 0.19 mmol), (4-hydroxyphenyl)boronic acid (31 mg, 0.23 mmol), Cs_2_CO_3_ (123 mg, 0.38 mmol), Pd(OAc)_2_ (3 mg, 0.013 mmol), EtOH (1.2 mL), and water (0.4 mL). The reaction was finished after 30 min. The desired compound was obtained after purification by column chromatography (EtOAc/Hex, 1:8 *v*/*v*). Yield: 50 mg (67%), yellow-brown crystalline solid, mp = 199–200 °C, R*_f_* = 0.17 (EtOAc/Hex, 1:3 *v*/*v*). ^1^H NMR (700 MHz, CDCl_3_): δ 1.50 (3H, t, *J* = 7.6 Hz, CH_3_), 2.05 (1H, s, OH), 3.21 (2H, q, *J* = 7.6 Hz, CH_2_), 6.58–6.64 (2H, m, C7Ph 3,5-H), 7.15–7.18 (1H, m, C6Ph C-4), 7.19–7.24 (4H, m, C6Ph 3,5-H and C7Ph 2,6-H), 7.40–7.45 (3H, m, C6Ph 2,6-H and NPh 4-H), 7.49–7.53 (2H, m, NPh 3,5-H), 7.85–7.90 (2H, m, NPh 2,6-H), 8.60 (1H, s, 3-H). ^13^C NMR (176 MHz, CDCl_3_): δ 13.6 (CH_3_), 30.1 (CH_2_), 115.3 (C7Ph C-3,5), 118.9 (C-3a), 120.7 (C-7), 121.6 (NPh C-2,6), 122.0 (C-3), 126.9 (C6Ph C-4), 127.3 (C7Ph C-1), 127.7 (C6Ph C-3,5), 128.6 (NPh C-4), 129.6 (NPh C-3,5), 130.6 (C6Ph C-2,6), 132.1 (C7Ph C-2,6), 140.0 (NPh C-1), 140.8 (C7Ph C-1), 148.9 (C-6), 152.1 (C-7a), 155.3 (C7Ph C-4), 159.0 (C-4). ^15^N NMR (71 MHz, CDCl_3_): δ −147.5 (N-2), −98.5 (N-1), −85.0 (N-5). IR (KBr, ν, cm^−1^): 3147 (OH), 3064 (CH_arom_), 2963, 2932, 2873 (CH_aliph_), 1613, 1586, 1507, 1481, 1267, 1209 (C=C, C=N, C–N), 1040 (C–O), 816, 760, 695 (CH=CH of mono- and disubstituted benzenes). MS (ES^+^): *m/z* (%): 391 ([M + H]^+^, 98). HRMS (ESI) for C_26_H_22_N_3_O ([M + H]^+^): requires 392.1757 and found 392.1757.

##### 4-Isopropyl-7-(4-methoxyphenyl)-4-methyl-2,6-diphenyl-2*H*-pyrazolo[4,3-*c*]pyridine **38**

4-Isopropyl-7-(4-methoxyphenyl)-4-methyl-2,6-diphenyl-2*H*-pyrazolo[4,3-*c*]pyridine **38** was prepared in accordance with general procedure (D) from 7-iodo-4-isopropyl-2,6-diphenyl-2*H*-pyrazolo[4,3-*c*]pyridine **16** (100 mg, 0.23 mmol), 4-methoxyphenyl)boronic acid (42 mg, 0.27 mmol), Cs_2_CO_3_ (148 mg, 0.46 mmol), Pd(OAc)_2_ (4 mg, 0.015 mmol), EtOH (1.5 mL), and water (0.5 mL). The reaction was finished after 1 h. The desired compound was obtained after purification by column chromatography (EtOAc/Hex, 1:8 *v*/*v*). Yield: 80 mg (83%), white crystalline solid, mp = 159–160 °C, R*_f_* = 0.54 (EtOAc/Hex, 1:3 *v*/*v*). ^1^H NMR (700 MHz, CDCl_3_): δ 1.55 (6H, d, *J* = 7.0 Hz, CH-(C*H*_3_)_2_), 3.53 (1H, p, *J* = 7.0 Hz, C*H*-(CH_3_)_2_), 3.83 (3H, s, OCH_3_), 6.86–6.91 (2H, m, C7Ph 3,5-H), 7.20–7.24 (1H, m, C6Ph 4-H), 7.24–7.28 (2H, m, C6Ph 3,5-H), 7.39–7.45 (3H, m, C7Ph 2,6-H and NPh 4-H), 7.50–7.55 (4H, m, C6Ph 2,6-H and NPh 3,5-H), 7.89–7.94 (2H, m, NPh 2,6-H), 8.63 (1H, s, 3-H). ^13^C NMR (176 MHz, CDCl_3_): δ 22.0 (CH-(*C*H_3_*)*_2_), 36.2 (C*H*-(CH_3_)_2_), 55.3 (OCH_3_), 113.7 (C7Ph C-3,5), 118.1 (C-3a), 120.3 (C-7), 121.2 (C-3), 121.5 (NPh C-2,6), 127.0 (C6Ph C-4), 127.8 (C6Ph C-3,5), 128.5 (NPh C-4), 128.6 (C7Ph C-1), 129.7 (NPh C-3,5), 131.0 (C6Ph C-2,6), 132.4 (C7Ph C-2,6), 140.4 (NPh C-1), 141.4 (C6Ph C-1), 148.5 (C-6), 152.6 (C-7a), 158.7 (C7Ph C-4), 162.5 (C-4). ^15^N NMR (71 MHz, CDCl_3_): δ −147.9 (N-2), −96.8 (N-1), −83.9 (N-5). IR (ν, cm^−1^): 3136, 3041 (CH_arom_), 2961, 2926, 2869 (CH_aliph_), 1587, 1547, 1506, 1482, 1464, 1379, 1288, 1242 (C=C, C=N, C–N), 1212, 1177, 1031 (C-O-C), 763, 758, 689 (CH=CH of mono- and disubstituted benzenes). MS (ES^+^): *m/z* (%): 421 ([M + 2H]^+^, 97.4). HRMS (ESI) for C_28_H_26_N_3_O ([M + H]^+^): requires 420.2070 and found 420.2070.

##### 7-(4-Methoxyphenyl)-2,4,6-triphenyl-2*H*-pyrazolo[4,3-*c*]pyridine **39**

7-(4-Methoxyphenyl)-2,4,6-triphenyl-2*H*-pyrazolo[4,3-*c*]pyridine **39** was prepared in accordance with general procedure (D) from 7-iodo-2,4,6-triphenyl-2*H*-pyrazolo[4,3-*c*]pyridine **17** (64 mg, 0.135 mmol), (4-methoxyphenyl)boronic acid (25 mg, 0.16 mmol), Cs_2_CO_3_ (88 mg, 0.27 mmol), Pd(OAc)_2_ (2 mg, 0.009 mmol), EtOH (0.9 mL), and water (0.3 mL). The reaction was finished after 45 min. The desired compound was obtained after purification by column chromatography (EtOAc/Hex, 1:10 *v*/*v*). Yield: 38 mg (62%), yellow crystalline solid, mp = 246–247 °C, R*_f_* = 0.54 (EtOAc/Hex, 1:3 *v*/*v*). ^1^H NMR (700 MHz, CDCl_3_): δ 3.84 (3H, s, CH_3_), 6.88-6.92 (2H, m, C7Ph C-3,5), 7.23–7.26 (1H, m, C6Ph 4-H), 7.26–7.31 (2H, m, 3,5-H), 7.41–7.44 (1H, m, NPh 4-H), 7.48-7.54 (5H, m, C4Ph 4-H; C7Ph 2,6-H; NPh 3,5-H), 7.55–7.62 (4H, m, C6Ph 2,6-H; C4Ph 3,5-H), 7.91–7.96 (2H, m, NPh 2,6-H), 8.14–8.20 (2H, m, C4Ph 2,6-H), 8.80 (1H, s, 3-H). ^13^C NMR (176 MHz, CDCl_3_): δ 55.2 (CH_3_), 113.5 (C7Ph C-3,5), 118.1 (C-3a), 121.2 (C-7), 121.4 (NPh C-2,6), 122.3 (C-4), 127.0 (C6Ph C-4), 127.7 (C6Ph C-3,5), 128.1 (C7Ph C-1), 128.3 (C4Ph C-2,6), 128.6 (NPh C-4), 128.8 (C4Ph C-3,5), 129.4 (C4Ph C-4), 129.6 (NPh C-3,5), 130.6 (C6Ph C-2,6), 132.4 (C7Ph C-2,6), 139.7 (C4Ph C-1), 140.1 (NPh C-1), 141.1 (C6Ph C-1), 149.2 (C-6), 152.8 (C-7a), 153.4 (C-4), 158.8 (C7Ph C-4). ^15^N NMR (71 MHz, CDCl_3_): δ −145.6 (N-2), −96.6 (N-1), N-5 not found. IR (KBr, ν, cm^−1^): 3056 (CH_arom_), 2924, 2833 (CH_aliph_), 1609, 1504, 1463, 1353, 1249, 1175 (C=C, C=N, C–N), 1038 (C–O), 838, 754, 695, 687 (CH=CH of mono- and disubstituted benzenes). MS (ES^+^): *m/z* (%): 454 ([M + H]^+^, 96). HRMS (ESI) for C_31_H_24_N_3_O ([M + H]^+^): requires 454.1914 and found 454.1914.

#### 3.2.6. General Procedure (E) of 4-(4-Ethyl-2,6-diphenyl-2*H*-pyrazolo[4,3-*c*]pyridin-7-yl)phenol **37** Alkylation 

4-(4-Ethyl-2,6-diphenyl-2*H*-pyrazolo[4,3-*c*]pyridin-7-yl)phenol **37** (1 equivalent) was dissolved in DMF. Then, NaH (60% in mineral oil) (1.1 equivalents) was added at room temperature. Then, an appropriate amount of alkyl iodide (1.1 equivalents) was added at 70 °C and the mixture was stirred for 1 h. Upon completion (monitored by TLC), the reaction mixture was cooled to room temperature, diluted with water (20 mL), and extracted with EtOAc (3 × 25 mL). The combined organic layers were washed with brine (10 mL), dried over anhydrous Na_2_SO_4_, filtered, and evaporated under reduced pressure. The residue was purified by column chromatography.

##### 7-(4-Ethoxyphenyl)-4-ethyl-2,6-diphenyl-2*H*-pyrazolo[4,3-*c*]pyridine **40**

7-(4-Ethoxyphenyl)-4-ethyl-2,6-diphenyl-2*H*-pyrazolo[4,3-*c*]pyridine **40** was prepared in accordance with general procedure (E) from (4-ethyl-2,6-diphenyl-2*H*-pyrazolo[4,3-*c*]pyridin-7-yl)phenol **37** (60 mg, 0.15 mmol), NaH (60%) (7 mg, 0.17 mmol), ethyl iodide (0.014 mL, 0.17 mmol), and DMF (2 mL). The desired compound was obtained after purification by column chromatography (EtOAc/Hex, 1:4 *v*/*v*). Yield: 62 mg (97%), yellow crystalline solid, mp = 140–141 °C, R*_f_* = 0.38 (EtOAc/Hex, 1:3 *v*/*v*). ^1^H NMR (700 MHz, CDCl_3_): δ 1.42 (3H, t, *J* = 7.0 Hz, OCH_2_C*H*_3_), 1.52 (3H, t, *J* = 7.6 Hz, CH_2_C*H*_3_), 3.20 (2H, q, *J* = 7.6 Hz, C*H*_2_CH_3_), 4.04 (2H, q, OC*H*_2_CH_3_), 6.82–6.88 (2H, m, C7Ph 3,5-H), 7.19–7.23 (1H, m, C6Ph 4-H), 7.23–7.27 (2H, m, C6Ph 3,5-H), 7.37–7.43 (3H, m, NPh 4-H and C6Ph 2,6-H), 7.47–7.52 (4H, m, NPh 3,5-H and C7Ph 2,6-H), 7.88–7.93 (2H, m, NPh 2,6-H), 8.60 (1H, s, 3-H). ^13^C NMR (176 MHz, CDCl_3_): δ 13.4 (CH_2_*C*H_3_), 14.9 (OCH_2_*C*H_3_), 30.3 (*C*H_2_CH_3_), 63.2 (O*C*H_2_CH_3_), 114.0 (C7Ph C-3,5), 119.0 (C-3a), 120.4 (C-7), 121.2 (C-3), 121.3 (NPh C-2,6), 126.8 (C6Ph C-4), 127.7 (C6Ph C-3,5), 128.1 (C7Ph C-1), 128.4 (NPh C-4), 129.5 (NPh C-3,5), 130.6 (C6Ph C-2,6), 132.3 (C7Ph C-2,6), 140.1 (NPh C-1), 141.2 (C6Ph C-1), 148.8 (C-6), 152.0 (C-7a), 158.0 (C7Ph C-4), 158.8 (C-4). ^15^N NMR (71 MHz, CDCl_3_): δ −147.6 (N-2), −96.5 (N-1), −83.8 (N-5). IR (KBr, ν, cm^−1^): 3059 (CH_arom_), 2983, 2930 (CH_aliph_), 1587, 1506, 1479, 1375, 1244, 1179 (C=C, C=N, C–N), 1047 (C–O), 826, 754, 670, 686 (CH=CH of mono- and disubstituted benzenes). MS (ES^+^): *m/z* (%): 419 ([M + H]^+^, 97). HRMS (ESI) for C_28_H_26_N_3_O ([M + H]^+^): requires 420.2070 and found 420.2070.

##### 4-Ethyl-2,6-diphenyl-7-(4-propoxyphenyl)-2*H*-pyrazolo[4,3-*c*]pyridine **41**

4-Ethyl-2,6-diphenyl-7-(4-propoxyphenyl)-2*H*-pyrazolo[4,3-*c*]pyridine **41** was prepared in accordance with general procedure (E) from (4-ethyl-2,6-diphenyl-2*H*-pyrazolo[4,3-*c*]pyridin-7-yl)phenol **37** (60 mg, 0.15 mmol), NaH (60%) (7 mg, 0.17 mmol), 1-iodo propane (0.016 mL, 0.17 mmol), and DMF (2 mL). The desired compound was obtained after purification by column chromatography (EtOAc/Hex, 1:6 *v*/*v*). Yield: 64 mg (96%), yellow crystalline solid, mp = 117–118 °C, R*_f_* = 0.41 (EtOAc/Hex, 1:3 *v*/*v*). ^1^H NMR (700 MHz, CDCl_3_): δ 1.04 (3H, t, *J* = 7.4 Hz, OCH_2_CHC*H*_3_), 1.52 (3H, t, *J* = 7.6 Hz, CH_2_C*H*_3_), 1.81 (2H, hept, *J* = 7.1 Hz, OCH_2_C*H*_2_CH_3_), 3.20 (2H, q, *J* = 7.6 Hz, C*H*_2_CH_3_), 3.92 (2H, t, *J* = 6.6 Hz, OC*H*_2_CHCH_3_), 6.83–6.87 (2H, m, C7Ph 3,5-H), 7.19–7.22 (1H, m, C6Ph 4-H), 7.23–7.27 (2H, m, C6Ph 3,5-H), 7.37–7.42 (3H, m, C7Ph 2,6-H and NPh 4-H), 7.47–7.51 (4H, m, NPh 3,5-H and C6Ph 2,6-H), 7.88–7.92 (2H, m, NPh 2,6-H), 8.59 (1H, s, 3-H). ^13^C NMR (176 MHz, CDCl_3_): δ 10.5 (OCH_2_CH*C*H_3_), 13.4 (CH_2_*C*H_3_), 22.6 (OCH_2_*C*HCH_3_), 30.3 (*C*H_2_CH_3_), 69.3 (O*C*H_2_CHCH_3_), 114.0 (C7Ph C-3,5), 119.0 (C-3a), 120.4 (C-7), 121.2 (C-3), 121.3 (NPh C-2,6), 126.8 (C6Ph C-4), 127.7 (C6Ph C-3,5), 128.0 (C7Ph C-1), 128.3 (NPh C-4), 129.5 (NPh C-3,5), 130.6 (C6Ph C-2,6), 132.3 (C7Ph C-2,6), 140.1 (NPh C-1), 141.2 (C6Ph C-1), 148.8 (C-6), 152.0 (C-7a), 158.2 (C7Ph C-4), 158.8 (C-4). ^15^N NMR (71 MHz, CDCl_3_): *δ* −147.5 (N-2), −96.2 (N-1), −83.5 (N-5). IR (KBr, ν, cm^−1^): 3045 (CH_arom_), 2970, 2931, 2872 (CH_aliph_), 1609, 1588, 1508, 1250, 1242, 1176 (C=C, C=N, C–N), 1041 (C–O), 758, 697, 687 (CH=CH of mono- and disubstituted benzenes). MS (ES^+^): *m/z* (%): 433 ([M + H]^+^, 95). HRMS (ESI) for C_29_H_28_N_3_O ([M + H]^+^): requires 434.2227 and found 434.2227.

##### 4-Ethyl-7-(4-isopropoxyphenyl)-2,6-diphenyl-2*H*-pyrazolo[4,3-*c*]pyridine **42**

4-Ethyl-7-(4-isopropoxyphenyl)-2,6-diphenyl-2*H*-pyrazolo[4,3-*c*]pyridine **42** was prepared in accordance with general procedure (E) from (4-ethyl-2,6-diphenyl-2*H*-pyrazolo[4,3-*c*]pyridin-7-yl)phenol **37** (60 mg, 0.15 mmol), NaH (60%) (7 mg, 0.17 mmol), 2-iodo propane (0.016 mL, 0.17 mmol), and DMF (2 mL). The desired compound was obtained after purification by column chromatography (EtOAc/Hex, 1:6 *v*/*v*). Yield: 52 mg (80%), yellow crystalline solid, mp = 139–140 °C, R*_f_* = 0.41 (EtOAc/Hex, 1:3 *v*/*v*). ^1^H NMR (700 MHz, CDCl_3_): δ 1.35 (6H, d, *J* = 6.1 Hz, CH(C*H*_3_)_2_), 1.53 (3H, t, *J* = 7.6 Hz, CH_2_C*H*_3_), 3.21 (2H, q, *J* = 7.6 Hz, C*H*_2_CH_3_), 4.56 (1H, hept, *J* = 6.1 Hz, CH), 6.82–6.86 (2H, m, C7Ph 3,5-H), 7.20–7.23 (1H, m, C6Ph 4-H), 7.24–7.27 (2H, m, C6Ph 3,5-H), 7.38–7.43 (3H, m, NPh 4-H and C67-Ph 2,6-H), 7.48–7.53 (4H, m, C6Ph 2,6-H and NPh 3,5-H), 7.89–7.94 (2H, m, NPh 2,6-H), 8.61 (1H, s, 3-H). ^13^C NMR (176 MHz, CDCl_3_): δ 13.4 (CH_2_*C*H_3_), 22.1 (CH(*C*H_3_)_2_), 30.3 (*C*H_2_CH_3_), 69.7 (OCH_2_), 115.3 (C7Ph C-3,5), 119.0 (C-3a), 120.4 (C-7), 121.25 (C-3), 121.29 (NPh C-2,6), 126.8 (C6Ph C-4), 127.7 (C6Ph C-3,5), 127.9 (C7Ph C-1), 128.4 (NPh C-4), 129.5 (NPh C-3,5), 130.6 (C6Ph C-2,6), 132.3 (C7Ph C-2,6), 140.2 (NPh C-1), 141.2 (C6Ph C-1), 148.8 (C-6), 152.0 (C-7a), 156.9 (C7Ph C-4), 158.8 (C-4). ^15^N NMR (71 MHz, CDCl_3_): δ −147.5 (N-2), −96.2 (N-1), −83.6 (N-5). IR (KBr, ν, cm^−1^): 3124, 3063 (CH_arom_), 2971, 2933 (CH_aliph_), 1608, 1588, 1507, 1280, 1238, 1182 (C=C, C=N, C–N), 1036 (C–O), 765, 702, 694 (CH=CH of mono- and disubstituted benzenes). MS (ES^+^): *m/z* (%): 433 ([M + H]^+^, 97). HRMS (ESI) for C_29_H_28_N_3_O ([M + H]^+^): requires 434.2227 and found 434.2227.

### 3.3. Optical Properties

The UV–vis spectra of 10^−4^ mol solutions of the compounds in THF were recorded on a Shimadzu 2600 UV/vis spectrometer. The fluorescence spectra were recorded on an FL920 fluorescence spectrometer from Edinburgh Instruments. The PL quantum yields (*Φ_f_*) were measured from dilute THF solutions by an absolute method using the Edinburgh Instruments integrating sphere excited with a Xe lamp. The optical densities of the sample solutions were ensured to be below 0.1 to avoid reabsorption effects. All optical measurements were performed at room temperature under ambient conditions.

A Britton–Robinson buffer (a solution consisting of 0.04 M H_3_PO_4_, 0.04 M CH_3_COOH, and 0.04 M H_3_BO_3_) was used to evaluate the pH dependence of the spectral characteristics of the compounds. The final pH values of the solutions were adjusted by 0.2 M NaOH.Stock solutions (4 mM) of the compounds were prepared in DMSO and further diluted in a Britton–Robinson buffer to a final concentration of 2 μM for spectroscopic analyses. Absorption spectra at pH 5, 7, and 9 for all compounds and in the 2–11 pH range with 0.5 step for selected compounds were measured using a Specord 250 Plus spectrophotometer in appropriate Britton–Robinson buffers. The spectra were measured in the 240–450 nm interval with a step of 1 nm, a 1 nm bandpass, and an integration time of 0.5 s. The samples were placed into a quartz cuvette with an optical path of 1 cm. The baseline was measured for the cuvette containing the solvent only.The steady-state excitation and emission spectra of 2 μM solutions of all the compounds at pH 5, 7, and 9 and in the 2–11 pH range with a 0.5 step for selected compounds were recorded on a Fluorolog-3 fluorimeter in the quartz cuvette with the 1 cm optical path (both in excitation and emission). Bandpasses in both the excitation and emission monochromator were set to 2 nm, and the spectra were scanned with the 1 nm step and an integration time 0.2 s per data point at 22 °C. Emission spectra were recorded in a 370–700 nm range with excitation at 360 nm.The quantum yield was estimated via integration of the fluorescence intensity over a range of 370–700 nm, and a 2.5 μM quinine sulphate solution in 0.05 M H_2_SO_4_ was used as a standard (*Φ_f_* = 60%) [76].

### 3.4. Biology

#### 3.4.1. Cell Cultures

Human cell lines were obtained from European Collection of Authenticated Cell Cultures (K562, MCF-7) or Cell Lines Service (MV4-11), and they were cultivated according to the provider’s instructions. Briefly, the MCF-7 and K562 cell lines were maintained in a DMEM medium (Sigma-Aldrich, St. Louis, MO, USA), and the MV4-11 cell line was maintained in an RPMI-1640 medium. All media were supplemented with 10% foetal bovine serum (Biowest, Nuaillé, France), penicillin (100 U/mL; Sigma-Aldrich, St. Louis, MO, USA), and streptomycin (100 mg/mL; Sigma-Aldrich, St. Louis, MO, USA), and cells were cultivated at 37 °C in 5% CO_2_.

#### 3.4.2. Antiproliferative Activity Assay

Cells were treated in triplicate with six different doses of each compound for 72 h. After treatment, an MTT solution (Sigma-Aldrich, St. Louis, MO, USA) was added for 4 h, the formazan was subsequently dissolved by adding a 10% SDS solution (Sigma-Aldrich, St. Louis, USA), and absorbance was measured at 570 nm using a Tecan M200Pro microplate reader (Biotek, Winooski, VT, USA). The GI_50_ value, the drug concentration lethal to 50% of the cells, was calculated from the dose–response curves. Flavopiridol (MedChemExpress, Monmouth Junction, NJ, USA) was used as a reference drug.

#### 3.4.3. Immunoblotting

After the treatment of the K562 cells, lysates in a RIPA buffer were prepared and proteins were separated on SDS-polyacrylamide gels and electroblotted onto nitrocellulose membranes. After blocking, overnight incubation with specific primary antibodies, and incubation with peroxidase-conjugated secondary antibodies, the peroxidase activity was detected with SuperSignal West Pico reagents (Thermo Scientific, Waltham, MA, USA) using a CCD camera LAS-4000 (Fujifilm, Tokyo, Japan). All primary antibodies were diluted in TBS containing 4% BSA and 0.1% Tween 20. The specific antibodies were purchased from Cell Signalling (Danvers, MA, USA; anti-PARP-1, clone 46D11; anti-cleaved caspase 9, clone E5Z7N; HRP-linked secondary antibodies), Sigma-Aldrich (St. Louis, MO, USA; anti-LC3B), and Santa Cruz Biotechnology (Dallas, TX, USA; anti-β-Actin, clone C4), or they were kindly gifted by dr. B. Vojtěšek (Masaryk Memorial Cancer Institute, Brno, Czech Republic; anti-PCNA, clone PC-10).

#### 3.4.4. Flow Cytometry

Asynchronously growing K562 cells were treated with a 10 µM concentration of test compounds for 24, 48, and 72 h, and 30 min before the end of incubation, the cells were labelled with 10 µM BrdU (Sigma-Aldrich, St. Loius, MO, USA) for 30 min. Subsequently, the cells were washed in PBS, fixed with ice-cold 70% ethanol, and denatured in 2 M HCl. After neutralization, the cells were stained with an anti-BrdU FITC-labelled antibody (eBioscience, San Diego, CA, USA) and propidium iodide (Sigma-Aldrich, St. Loius, MO, USA). Samples were then analysed by flow cytometry using a 488 nm laser (BD FACS Verse with software BD FACSuite™, version 1.0.6.; BD, Franklin Lakes, NJ, USA).

## 4. Conclusions

An efficient synthesis of 2,4,6,7-tetrasubstituted-2*H*-pyrazolo[4,3-*c*]pyridine derivatives was developed starting from easily accessible 1-phenyl-3-(2-phenylethynyl)-1*H*-pyrazole-4-carbaldehyde. The obtained compounds were evaluated for their antiproliferative activity against three cancer cell lines. Out of them, 4-(2,6-diphenyl-2*H*-pyrazolo[4,3-*c*]pyridin-7-yl)phenol **23** proved to be the most active, and further experiments revealed that it blocks proliferation and induces cell death in K562 cells. Moreover, the majority of the compounds were revealed to be pH-sensitive, and 7-(4-methoxyphenyl)-2,6-diphenyl-2*H*-pyrazolo[4,3-*c*]pyridine was found out to enable both fluorescence-intensity-based and ratiometric pH sensing.

## Data Availability

The data that support the findings of this study are available upon request.

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
