# Peer review of "Synthesis and Antiproliferative Activity of 2,4,6,7-Tetrasubstituted-2H-pyrazolo[4,3-c]pyridines"

_molecules, 2021, doi:10.3390/molecules26216747_

Round 1
Reviewer 1 Report
The manuscript describes the synthesis of a several tetrasubstituted pyrazolopyridine derivatives from pyrazole-4-carbaldehyde containing alkyne by key steps such as iodine-mediated electrophilic cyclization and Suzuki-Miyaura cross-coupling. The work appears to be worthy and these compounds were evaluated for their antiproliferative activity against some human cancer cell lines along with supplementary information supports synthesis of the correct compounds. In general, the manuscript is well presented, but there are a few improvements that can be made. I support publication of the manuscript after these concerns have been made.
My comments, suggestions, and questions to authors:
- In particular, I recommend authors to include separate figure contain varieties of few biologically relevant scaffolds of these skeletons reported by other groups.
- The authors must include fluorescence aspects in the abstract and conclusion as well.
- No specific compound numbers should be given in the abstract(see the
last sentence of the abstract). - Page 4, line 133, 148, 157: it should be ---(Table S1, SI file), (Table S2, SI file), and (Figure S1B, SI file)
- Page 3 & 4, line 123, 130: include “i” in reagents and conditions or keep conditions in the schemes
- Preparation method of starting materials 1—2 with schemes are not given (should include in SI file with citations)
- Page 7, line 249: must be white crystalline solid instead of white crystals (similarly in all cases in the MS file).
- Page 7, line 270: better to change CH3MgCl instead of methylMgCl (similarly in all cases in the MS file).
- I can see that from NMR spectroscopic data there is a missing sharp singlet pyrazole “1H” (for compound 4) as well missing carbons from 13C compared with expected. Please, double-check NMR data with spectra in all cases throughout the MS file.
- There is a confusion in 1H data of compound 5, ppm value 29 (1H, S) is it compound peak??? Some places they did not included (compound 6) and double-check NMR data with spectra in all cases throughout the MS file.
- I recommend authors to include the 19F-NMR (data and spectra) for fluorinated frameworks.
- Traces of impurities are visible on some of the NMR spectrums (ex. 37, 43 etc.…)
Reviewer 2 Report
In this manuscript, a novel series of 2,4,6,7-Tetrasubstituted-2H-pyrazolo[4,3-c]pyridines (13-42) were synthesized and characterized by 1H NMR, 13C NMR and ESI-HRMS spectrometry. All compounds were screened for their anti-proliferative activities against K562, MV4-11 and MCF-7 cancer cell lines. In vitro biological investigations revealed that most of compounds were active against studied cancer cell lines. Among tested compounds, compound 23 was found to be more potent.
Overall the manuscript is rich and interesting; and the paper structure is well-knit and suitable for publication in the journal, after minor revisions. The comments are listed as the following points:
- In abstract, lines 24-26, the following sentence “The most potent compounds displayed low micromolar GI50 values; and 4-hydoxyphenyl substituent at 7-position bearing, 4-unsubstituted derivative 23 proved to be the most active” should be reworded.
- “In current medicinal chemistry, incorporation of pyrazole nucleus serves as a common practice to develop new drug-like molecules with anti-cancer, anti-diabetic, anti-viral, anti-inflammatory, anti-bacterial, anti-fungal, anti-neurodegenerative, anti-tubercular, anthelmintic, antimalarial, photosentisizing properties.” The following reference should be given: “Molecules 2018, 23(1), 134; https://doi.org/10.3390/molecules23010134.
- In page 5, Table 1, the anti-proliferative activity of compounds (18-42) should be compared to a reference drug.
- The authors should explain why they did not test compounds (13-17) for their anti-proliferative activity.
- In supporting information, there are no Mass Spectrums of compounds (3-17).
Reviewer 3 Report
In this article, the authors study the synthesis of a series of tetrasubstituted pyrazolo-pyridine derivatives. The synthesis starts with pyrazole-4-carbaldehyde (2) derivatives, and which are synthesized based on the method reported from previous literature from compound 1. The structure of the synthesized compounds is well characterized using NMR and MS and complete data are presented in the manuscript and SI. The authors screened the synthesized compounds against antiproliferative activity on three different cancer cell lines. Several compounds showed good to moderate GI50 values (50% maximal inhibition of cell proliferation) against these cells (Table 1, compounds 23, 22, 21, etc.). All of these are well presented in this manuscript and in my opinion, this article will be interesting to Molecules readers.
Author Response
We would like to thank the reviewer for kind evaluation of our Manuscript.